# Learning to Optimize for Mixed-Integer Non-linear Programming

## Abstract

Mixed-integer non-linear programs (MINLPs) arise in various domains, such as energy systems and transportation, but are notoriously difficult to solve. Recent advances in machine learning have led to remarkable successes in optimization tasks, an area broadly known as *learning to optimize*. This approach includes using predictive models to generate solutions for optimization problems with continuous decision variables, thereby avoiding the need for computationally expensive optimization algorithms. However, applying learning to MINLPs remains challenging primarily due to the presence of integer decision variables, which complicate gradient-based learning. To address this limitation, we propose two differentiable correction layers that generate integer outputs while preserving gradient information. Combined with a soft penalty for constraint violation, our framework can tackle both the integrality and non-linear constraints in a MINLP. Experiments on three problem classes with convex/non-convex objective/constraints and integer/mixed-integer variables show that the proposed learning-based approach consistently produces high-quality solutions for parametric MINLPs extremely quickly. As problem size increases, traditional exact solvers and heuristic methods struggle to find feasible solutions, whereas our approach continues to deliver reliable results. Our work extends the scope of learning-to-optimize to MINLP, paving the way for integrating integer constraints into deep learning models. Our code is available at https://anonymous.4open.science/r/L2O-MINLP/.

## 1 Introduction

Mixed-integer optimization is fundamental to a broad spectrum of real-world applications spanning problems in fields as diverse as pricing Kleinert et al. (2021), battery dispatch (Nazir & Almassalkhi, 2021), transportation (Schouwenaars et al., 2001), and optimal control (Marcucci & Tedrake, 2020). These problems involve discrete decisions, such as determining the number of items or the activation of generators, combined with complex non-linear system constraints. Mixed-integer *linear* programming (MILP) has been widely adopted due to its well-established solution techniques. However, many practical problems exhibit non-linear relationships, leading to mixed-integer non-linear programs (MINLPs). Unlike MILPs, where techniques such as branch-and-bound (Land & Doig, 2010), cutting planes (Gomory, 2010), and heuristics (Crama et al., 2005; Johnson & McGeoch, 1997) have matured, MINLPs require more complex approaches due to the combination of discrete variables and non-convex constraints and objective function. Standard methods include outer approximation (Fletcher & Leyffer, 1994), spatial branch-and-bound (Belotti et al., 2009), and decomposition techniques (Nowak, 2005), but these often struggle to scale to large problems.

Many applications demand that MINLPs be solved within a limited time budget, further complicating the picture. To overcome this, learning-to-optimize (L2O) methods offer a promising alternative by leveraging machine learning (ML) to enhance or even replace conventional optimization approaches. In particular, *end-to-end optimization* directly maps input instance parameters to solutions of optimization problems through a trained model (Kotary et al., 2021b; Chen et al., 2022a). By identifying patterns in a distribution of similar instances of the same optimization problem and predicting solutions accordingly, end-to-end optimization can bypass traditional, computationally intensive optimization methods, enabling faster computation and improved scalability.

Many real-world applications have stringent requirements on operational, physical, or safety constraints. Thus, recent research in machine learning has focused on the feasibility issue. While various strategies exist, such as embedding hard constraints into neural network architectures (Hendriks et al., 2020), using penalty terms in loss functions for soft constraints (Pathak et al., 2015; Jia et al., 2017), or projecting solutions onto feasible regions (Donti et al., 2021), these methods are not directly applicable for problems that involve integer decisions.

This work tackles, for the first time, the non-differentiability associated with predicting integer variables using a deep neural network, in conjunction with non-linear objective function and constraints. This challenge has been underexplored in learning-based methods due to the absence of useful gradient information. To that end, we propose two differentiable correction layers for rounding, allowing for gradient-based optimization of a neural network that generates high-quality integer solutions while maintaining feasibility. Our contributions are as follows:

- We initiate the study of the learning-to-optimize problem in MINLP for the first time in the literature, a paradigm that can enable efficient solution generation as problem parameters vary.

- We develop differentiable correction layers that perform soft rounding of neural network outputs into integer assignments to decision variables.

- We adopt a self-supervised approach that requires no labeled data for training, making our method efficient and scalable to large problem instances.

- We evaluate our methods on diverse problem benchmarks and show that they find high-quality solutions extremely fast even for large-scale instances where other methods fail.

## 2 RELATED WORK

**End-to-end optimization.** End-to-end optimization focuses on training machine learning models to predict the problem solutions, bypassing the need for computationally expensive solvers. One of the early approaches was proposed by Hopfield & Tank (1985), who used Hopfield networks to solve the traveling salesperson problem by incorporating a Lagrangian penalty for constraint feasibility. Similarly, Fioretto et al. (2020) applied the Lagrangian penalty in the context of continuous non-linear optimization for energy systems. In addition to penalty-based methods for ensuring feasibility, Pan et al. (2020) embedded certain constraints directly into neural networks by leveraging the range of output values and solving linear systems. Although these supervised learning methods significantly reduce inference time, they typically require large offline datasets of solutions (Gleixner et al., 2021; Kotary et al., 2021a), which can be impractical for large-scale problems where generating solutions is computationally expensive. This limitation highlights the need for self-supervised learning approaches (Donti et al., 2021), which minimize both the objective function and constraint violation from the predicted values, without relying on the imimitation of pre-solved solutions. Our method first extends this self-supervised paradigm to problems involving discrete decision variables, further broadening its applicability to mixed-integer optimization.

**Constrained neural architectures.** Specific neural network architectures can be designed to impose certain classes of hard constraints. For instance, Hendriks et al. (2020) incorporate linear operator constraints directly into the model design. Vinyals et al. (2015) and Dai et al. (2017) leveraged the inherent structure of graphs to construct feasible solutions for the traveling salesperson problem. Additionally, Kervadec et al. (2022) demonstrated that employing a log-barrier method for inequality constraints improves accuracy, constraint satisfaction, and training stability. Penalty methods (Pathak et al., 2015; Jia et al., 2017), which impose inequality constraints through regularization terms in the loss function, have also gained popularity for constraining neural networks. As noted by Márquez-Neila et al. (2017), in practice, methods that incorporate hard constraints rarely outperform their soft constraint counterparts, despite the latter offering weaker theoretical performance guarantees. Building on penalty methods, Donti et al. (2021) proposed a differentiable correction approach to complete partial solutions for linear equations and project solutions onto the feasible region. In this paper, we adopt a penalty method for handling constraints and introduce two novel differentiable rounding correction layers to guarantee the integrity of the solution.

**Learning for mixed-integer programming.** There has been significant interest in using ML to accelerate the solution of integer programs. The vast majority of the work in this space focuses on learning search strategies for exact MILP solvers. This includes parameter tuning (Xu et al., 2011), preprocessing (Berthold & Hendel, 2021), branching variable selection (Khalil et al., 2016; Alvarez et al., 2017; Gasse et al., 2019; Zarpellon et al., 2021), node selection (He et al., 2014), heuristic selection (Chmiela et al., 2021), and cut selection and generation (Deza & Khalil, 2023). Another line of research in ML-for-MILP relates to learning to generate integer solutions heuristically (Nair et al., 2020; Khalil et al., 2022; Ding et al., 2020; Sonnerat et al., 2021; Song et al., 2020; Bertsimas & Stellato, 2022; Huang et al., 2023; Ye et al.). We refer to the surveys of Bengio et al. (2021) and Zhang et al. (2023) for more details. In contrast, there has been much less work on MINLP. Illustrative examples include the work of Cauligi et al. (2021) who proposed a two-stage algorithm for quickly finding high-quality solutions for mixed-integer convex programs (MICPs), Baltean-Lugojan et al. (2019) who use supervised learning to select cuts for quadratic optimization, Nowak et al. (2018) who learn to solve quadratic assignment problems with graph networks, and Bonami et al. (2022) who use a classifier to decide on the linearization of mixed-integer quadratic problems. Most relevant to our method is the recently proposed SurCO approach of Ferber et al. (2023). They focus on mixed-integer problems with non-linear objective and linear constraints, learning to approximate the former with a linear function for a simpler heuristic optimization. Our approach differs from all of the above in its scope, addressing the most general class of MINLPs.

**Differentiable optimization.** A different category of methods integrates optimization solvers as layers within deep neural network architectures (Agrawal et al., 2019). These methods can handle various types of optimization problems, such as quadratic programs (Amos & Kolter, 2017; Sambharya et al., 2023), stochastic optimization (Donti et al., 2017), submodular optimization (Djolonga & Krause, 2017), and even integer linear programs (Wilder et al., 2019; Berthet et al., 2020; Pogančić et al., 2020). In these approaches, optimization algorithms or solvers are embedded within the neural network, allowing gradients of optimization solvers to be computed and propagated during back-propagation. King et al. (2024) shows how differentiable optimization can enhance the convergence of proximal operator algorithms via end-to-end learning of proximal metrics. However, as Tang & Khalil (2024) noted, training with a differentiable optimizer requires iteratively solving optimization throughout the training process, making the computational burden prohibitively expensive. In contrast, our self-supervised approach generates solutions directly through neural network structures, eliminating the need to repeatedly call high-complexity solvers and thus significantly reducing computational overhead.

## 3    LEARNING TO OPTIMIZE MINLPS: A PROBLEM FORMULATION

A generic learning-to-optimize formulation for parametric mixed-integer non-linear programming is given by:

$$\min_{\Theta} \quad \mathbb{E}\big[\mathbf{f}(\hat{\mathbf{x}}, \boldsymbol{\xi})\big], \quad \text{s.t.} \quad \mathbf{g}(\hat{\mathbf{x}}, \boldsymbol{\xi}) \leq 0, \quad \hat{\mathbf{x}} \in \mathbb{R}^{n_r} \times \mathbb{Z}^{n_z}, \quad \hat{\mathbf{x}} = \boldsymbol{\psi}_{\Theta}(\boldsymbol{\xi}).$$

Here, $\boldsymbol{\xi}^i \in \mathbb{R}^{n_\xi}$ is a vector of instance parameters which vary across different instances; the mapping $\boldsymbol{\psi}_{\Theta}(\boldsymbol{\xi}^i)$ is a neural network with weights $\Theta$ that outputs a parametric solution $\hat{\mathbf{x}}^i$; $\hat{\mathbf{x}}^i = (\hat{\mathbf{x}}_r^i, \hat{\mathbf{x}}_z^i)$ is a predicted assignment for the mixed-integer decision variables, where $\hat{\mathbf{x}}_r^i \in \mathbb{R}^{n_r}$ and $\hat{\mathbf{x}}_z^i \in \mathbb{Z}^{n_z}$ represent the continuous and integer parts, respectively. The goal is to find the neural network weights that minimize the expected objective function $\mathbf{f}(\hat{\mathbf{x}}, \boldsymbol{\xi})$ over the parameter distribution, subject to the constraints $\mathbf{g}(\hat{\mathbf{x}}, \boldsymbol{\xi}) \leq 0$. Note that $\mathbf{g}(\cdot)$ is a vector-valued function representing one or more inequality constraints. As is typical in MINLP, we assume that the objective and constraint functions are differentiable.

As is typical, we will train the neural network using empirical risk minimization on a sample of $m$ training instances. Then, the average value of the objective function $\mathbf{f}(\cdot)$ serves as a natural loss function. Our approach is *self-supervised* since the loss calculation does not require any labeled data. This is particularly appealing as computing optimal or even feasible solutions to a MINLP is, in general, extremely challenging. Solely minimizing the average objective is insufficient if the solutions violate the constraints. Therefore, similarly to Donti et al. (2021), we incorporate penalty terms into the loss function to account for constraint violations, enhancing the feasibility of the

solution and resulting in a soft-constrained empirical risk minimization loss function given:

$$\mathcal{L}(\Theta) = \frac{1}{m} \sum_{i=1}^{m} \left[ \mathbf{f}(\hat{\mathbf{x}}^i, \boldsymbol{\xi}^i) + \lambda \cdot \|\mathbf{g}(\hat{\mathbf{x}}^i, \boldsymbol{\xi}^i)_+\|_1 \right] \text{ with } \hat{\mathbf{x}}^i = \boldsymbol{\psi}_\Theta(\boldsymbol{\xi}^i), \tag{2}$$

where $\|(\cdot)_+\|_1$ ensures only the sum of positive constraint violations are penalized (implemented via a `ReLU` function), and $\lambda > 0$ is a penalty hyperparameter that balances the trade-off between minimizing the objective function and satisfying the constraints.

## 4  PRELIMINARIES: DIFFERENTIATING THROUGH DISCRETE OPERATIONS

**Straight-through Estimator.**    The Straight-through Estimator (STE) (Bengio et al., 2013) is a simple yet effective method for handling non-differentiable operations in neural networks. In our approaches, STE plays a crucial role in enabling backpropagation through discrete operations. During the forward pass, STE applies a (non-differentiable) discrete operation, such as rounding a variable up or down, binarizing it, or using an indicator function $\mathbb{I}(\cdot)$. However, in the backward pass, STE replaces the non-existent gradient of these discrete functions with soft approximations. For rounding operations, the gradient of the identity function is used during backpropagation, whereas for binarization or indicator functions, the gradient of the Sigmoid function is applied.

**Gumbel-Sigmoid Noise.**    Although the STE is effective for backpropagating through discrete decisions, it lacks the stochasticity that can improve model training. This is where the Gumbel-noise method (Jang et al., 2016) comes into play. Specifically, Gumbel noise perturbs the logits before applying the Sigmoid function, allowing for randomness in the binary decisions. After this, a hard binarization step is applied using the STE, ensuring that the final outputs are discrete binary values while retaining gradients for backpropagation. Further technical details can be found in Appendix A.

## 5  LEARNING TO OPTIMIZE MINLPs WITH CORRECTION LAYERS

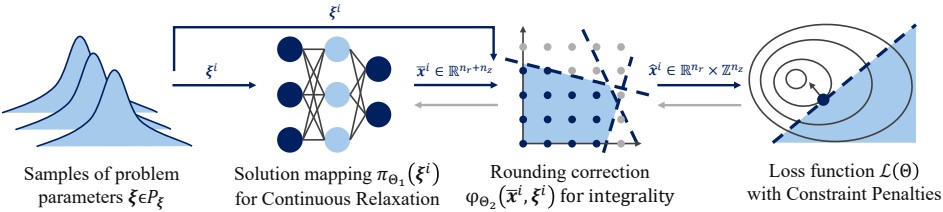

Figure 1: Conceptual diagram for our self-supervised differentiable programming-based solution approach for parametric MINLP problems.

Our learnable correction layers, Rounding Classification (RC) and Learnable Threshold (LT), are designed to handle the integrality constraints of MINLPs. We decompose the mapping $\psi_\Theta : \mathbb{R}^{n_\xi} \mapsto \mathbb{R}^{n_r} \times \mathbb{Z}^{n_z}$ from an instance parameter vector to a candidate mixed-integer solution into two steps:

1. The first step consists in applying a learnable *relaxed solution mapping* $\pi_{\Theta_1} : \mathbb{R}^{n_\xi} \mapsto \mathbb{R}^{n_r+n_z}$ encoded by a deep neural network with weights $\Theta_1$. It outputs a continuously relaxed solution $\bar{\mathbf{x}}^i \in \mathbb{R}^{n_r+n_z}$ without enforcing the integrality requirement. Note that continuous variables are also predicted in this first step.

2. The second step is a differentiable correction layer $\varphi_{\Theta_2} : \mathbb{R}^{n_r+n_z} \times \mathbb{R}^{n_\xi} \mapsto \mathbb{R}^{n_r} \times \mathbb{Z}^{n_z}$ that takes as input the instance parameter vector and the continuous solution produced in the first step, and outputs a candidate mixed-integer solution while maintaining differentiability. Here, $\Theta_2$ represents the weights of the neural network $\delta_{\Theta_2} : \mathbb{R}^{n_r+n_z} \times \mathbb{R}^{n_\xi} \mapsto \mathbb{R}^{n_r+n_z}$, which implicitly influences the rounding strategy employed by the correction layer $\varphi_{\Theta_2}$.

They differ in how to determine the rounding direction but are equally easy to train with gradient descent and fast at test time. RC utilizes a classification-based stochastic rounding approach, while LT employs learnable thresholds to determine rounding directions. Further details are provided in Appendix B.

**Alogorithm.** Algorithm 1 summarizes both of our approaches. Line 1 invokes the first step's network $\pi_{\Theta_1}$ and lines 2–11 describe both versions of $\varphi_{\Theta_2}$. Our correction layers are not only simple and efficiently computable but also designed to be trainable to refine its rounding strategy. While the STE and Gumbel-Sigmoid techniques have been used to train binarized or quantized neural networks, they have not been leveraged in the context of learning-to-optimize to our knowledge. As we will see in the experimental results, the simplicity of the correction layers is key to fast solution generation in large-scale MINLP problems.

---

**Algorithm 1** Learning-to-optimize MINLPs with Correction Layers: Forward Pass.

---

**Require:** Instance of the problem parameters $\boldsymbol{\xi}^i$, neural networks $\pi_{\Theta_1}(\cdot)$ and $\delta_{\Theta_2}(\cdot)$
 1: Predict a continuously relaxed solution $\bar{\mathbf{x}}^i \leftarrow \pi_{\Theta_1}(\boldsymbol{\xi}^i)$
 2: Obtain an initial correction prediction $\mathbf{h}^i \leftarrow \delta_{\Theta_2}(\bar{\mathbf{x}}^i, \boldsymbol{\xi}^i)$
 3: Update continuous variables: $\hat{\mathbf{x}}_r^i \leftarrow \bar{\mathbf{x}}_r^i + \mathbf{h}_r^i$
 4: Round integer variables down: $\hat{\mathbf{x}}_z^i \leftarrow \lfloor \bar{\mathbf{x}}_z^i \rfloor$
 5: **if** using *Rounding Classification* **then**
 6:     Compute $\mathbf{b}^i$ as the rounding direction using Gumbel-Sigmoid($\mathbf{h}_z^i$)
 7: **else if** using *Learnable Threshold* **then**
 8:     Compute $\mathbf{v}^i \in [0,1]^{n_z} \leftarrow$ Sigmoid($\mathbf{h}_z^i$)
 9:     Compute rounding direction: $\mathbf{b}^i \leftarrow \mathbb{I}\big((\bar{\mathbf{x}}_z^i - \hat{\mathbf{x}}_z^i) - \mathbf{v}^i > 0\big)$
10: **end if**
11: Update integer variables: $\hat{\mathbf{x}}_z^i \leftarrow \hat{\mathbf{x}}_z^i + \mathbf{b}^i$
12: **return** $\hat{\mathbf{x}}^i$

---

Finally, during training, the loss function eq. (2) is used to train the neural network weights $\Theta = \Theta_1 \cup \Theta_2$, implicitly taking into account the objective function value and constraint violations of the predicted mixed-integer solution $\hat{\mathbf{x}}^i$. This process is illustrated in Figure 1. Additionally, an example of the evolution of predicted solutions during training is provided in Appendix C for further visualization.

These approaches can be viewed as an end-to-end learnable version of the Relaxation Enforced Neighborhood Search (RENS) algorithm (Berthold, 2014). Instead of explicitly searching the neighborhood of the relaxed solution, the neural network implicitly learns the corrections required to achieve a feasible integer solution by exploring the solution space near the integer variables while updating the continuous variables.

# 6 EXPERIMENTAL RESULTS

## 6.1 EXPERIMENTAL SETUP

**Methods.** Table 1 provides an overview of all the methods used in the following experiments. A 1000-second time limit is enforced for all methods and problems. The experiments evaluate our learning-based methods, Rounding Classification (RC) and Learnable Threshold (LT), against traditional exact optimization (EX), which can compute optimal solutions but is often computationally expensive, and heuristic-based approaches such as Rounding after Relaxation (RR) and root node solutions (N1), which offer faster results without quality guarantees. Note that baselines EX and N1 include a wide range of heuristics that are embedded in the MINLP solver of choice (Gurobi or SCIP) and that are executed in conjunction with the tree search procedure; we are also implicitly comparing to these heuristics, not just to the exact search. As such, the competing methods cover a broad spectrum of optimization strategies, from exact solvers to fast heuristics, allowing for a comprehensive evaluation of solution quality and computational efficiency. In addition, we evaluate two ablation baselines, which isolate different aspects of our correction layers $\varphi_{\Theta_2}$ to highlight their impact on performance. Details of these ablation studies, including methodology and results, are provided in Appendix F.

**Problem classes.** We tested the methods on a variety of optimization problems, including integer convex quadratic problems, simple integer non-convex problems, and high-dimensional mixed-integer Rosenbrock problems. These problem classes were selected to cover both convex and non-convex scenarios and to evaluate the scalability of the methods in higher-dimensional settings. Each

Table 1: Summary of Methods. Methods with "*" use a trained model.

| Method | Abbr | Description |
|---|---|---|
| Rounding Classification* | RC* | Learning-based rounding approach using classification for integer variable rounding. |
| Learnable Threshold* | LT* | Learning-based method where a neural network learns the threshold for rounding integer variables. |
| Exact Optimization | EX | Solves the problem using Gurobi for convex problems and SCIP + Ipopt for non-convex problems. |
| Rounding after Relaxation | RR | Solves the continuous relaxation, then rounds the continuous solution to the nearest integer. |
| Root Node Solution | N1 | A feasible solution from the root node of the solver, which uses heuristics after continuous relaxation with cutting planes. |

method was assessed in terms of objective value, constraint violation, and solving time, providing a comprehensive view of their performance across different types of problems. In addition, we evaluated our methods on integer linear programs (MILPs), in which the dataset from the MIP Workshop 2023 Computational Competition Bolusani et al. (2023). These experiments primarily serve to demonstrate that our methods can also handle integer linear cases, though the use of MILP solvers may be preferable. Further details are provided in Appendix H.

**Training protocol.** The solution mapping $\pi_{\Theta_1}$ used across all learning-based methods (RC, LT, and RL) and the rounding correction network $\varphi_{\Theta_2}$ for RC and LT are based on fully connected layers with ReLU activations. Further details regarding the network hyperparameters can be found in Appendix D. For all problems, the training samples 8,000 instances from the distribution, and the test set includes 100 instances. An additional set of 1,000 instances was used for validation to fine-tune the models and select hyperparameters.

**Computational setup.** All experiments were conducted on a system with 2 Intel Silver 4216 Cascade Lake @ 2.1GHz CPUs, 64GB RAM, and 4 NVIDIA V100 Volta GPUs. The software environment was configured with Python 3.10.13, PyTorch 2.5.0+cu122 (Paszke et al., 2019) for deep learning models, and NeuroMANCER 1.5.2 (Drgona et al., 2023) for modeling parametric constrained optimization problems. Gurobi 11.0.1 (Gurobi Optimization, LLC, 2021) is used as the exact method for convex quadratic problems; beyond quadratic polynomials, Gurobi needs to approximate non-linearities using piecewise-linear functions. For those more general mixed-integer non-convex problems, we use SCIP 9.0.0 (Bestuzheva et al., 2021) with Ipopt 3.14.14 (Wächter & Biegler, 2006) as the continuous non-linear solver. Note that Gurobi and SCIP are considered to be among the state-of-the-art solvers for MINLP, as noted by Lundell & Kronqvist (2022) who performed a comprehensive benchmarking of more than ten MINLP solvers: "It is clear, however, that the global solvers Antigone, BARON, Couenne and SCIP are the most efficient at finding the correct primal solution when regarding the total time limit. [...] Gurobi also is very efficient when considering that it only supports a little over half of the total number of problems!"

**Overall results.** As shown in Figure 2, the exact solvers like Gurobi and SCIP gradually improve the objective value over time, but this often comes at a high computational cost. For more complex problems, they may even fail to find feasible solutions within reasonable time limits. In contrast, our methods, RC and LT, achieve high-quality, feasible solutions in mere milliseconds. Even when accounting for the 131.72 seconds required to train the neural network for the Rosenbrock problem, our approaches remain significantly more efficient. Once trained, these models generalize well to unseen instances, making them ideal for repeated problem-solving scenarios where the training cost is amortized. Additionally, RC and LT could provide high-quality initial solutions for exact solvers, reducing the search space and accelerating convergence, thus enhancing the performance of traditional optimization methods.

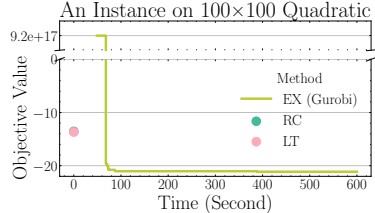 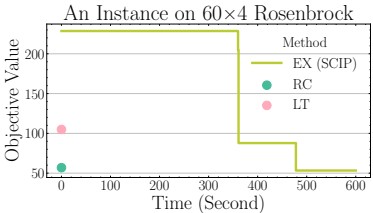

Figure 2: Illustration of objective value evolution for a $100 \times 100$ Convex Quadratic instance and $60 \times 4$ Rosenbrock instance over 600 seconds.

Table 2: Result for a Convex Quadratic Problem. Each problem size is evaluated on a test set of 100 instances. "Obj Mean" and "Obj Median" represent the mean and median objective values for this minimization problem, with smaller values being better. "% Infeasible" denotes the fraction of infeasible solutions, and "Time (Sec)" is the average solving/inference time per instance. The "—" symbol indicates that no solution is found for any instance within 1000 seconds.

| Method | Metric | 5×5 | 10×10 | 20×20 | 50×50 | 100×100 | 200×200 | 500×500 | 1000×1000 |
|---|---|---|---|---|---|---|---|---|---|
| RC | Obj Mean | 0.827 | −0.773 | −3.859 | −11.967 | −12.809 | −29.830 | −67.518 | −129.088 |
| | Obj Median | 0.641 | −1.180 | −3.888 | −12.013 | −12.760 | −29.965 | −67.633 | −129.263 |
| | % Infeasible | 0% | 1% | 0% | 1% | 1% | 2% | 0% | 0% |
| | Time (Sec) | 0.0019 | 0.0019 | 0.0021 | 0.0019 | 0.0021 | 0.0025 | 0.0026 | 0.0045 |
| LT | Obj Mean | 0.869 | −1.419 | −3.830 | −11.606 | −12.227 | −30.688 | −69.136 | −120.480 |
| | Obj Median | 0.672 | −1.702 | −3.818 | −11.736 | −12.299 | −31.030 | −69.222 | −120.651 |
| | % Infeasible | 0% | 0% | 0% | 0% | 0% | 7% | 0% | 5% |
| | Time (Sec) | 0.0019 | 0.0019 | 0.0019 | 0.0019 | 0.0024 | 0.0026 | 0.0028 | 0.0047 |
| EX | Obj Mean | 0.294 | −2.779 | −5.120 | −15.928 | −20.790 | — | — | — |
| | Obj Median | 0.129 | −2.991 | −5.130 | −15.956 | −20.778 | — | — | — |
| | % Infeasible | 0% | 0% | 0% | 0% | 0% | — | — | — |
| | Time (Sec) | 0.496 | 0.664 | 8.728 | 1520.73 | 1237.53 | — | — | — |
| RR | Obj Mean | 0.211 | −2.858 | −5.179 | −16.173 | −21.922 | −46.727 | −106.526 | −213.312 |
| | Obj Median | 0.058 | −3.033 | −5.217 | −16.205 | −21.892 | −46.755 | −106.536 | −213.292 |
| | % Infeasible | 97% | 100% | 100% | 100% | 100% | 100% | 100% | 100% |
| | Time (Sec) | 0.411 | 0.412 | 0.417 | 0.440 | 0.583 | 0.846 | 2.639 | 8.874 |
| N1 | Obj Mean | 0.549 | 1.2e15 | 9.8e07 | 1.7e17 | 1.5e18 | — | — | — |
| | Obj Median | 0.369 | −1.900 | 9.600 | 2.4e17 | 1.4e18 | — | — | — |
| | % Infeasible | 0% | 0% | 0% | 0% | 0% | — | — | — |
| | Time (Sec) | 0.420 | 0.422 | 0.415 | 0.498 | 104.204 | — | — | — |

## 6.2 CONVEX QUADRATIC PROBLEM

Since there is a lack of publicly available datasets for parametric MINLPs, the convex quadratic problems used in the experiments are adapted from Donti et al. (2021), which originally focused on learning under continuous constraints. We introduced integrality constraints on all decision variables to tailor these problems to our discrete setting. Additionally, we removed equality constraints to avoid the issue of generating infeasible instances. These modifications ensure compatibility with our framework while preserving the essential structure of the original problems. Further details on the mathematical formulation and data generation process can be found in Appendix E.

We experimented with quadratic problems of different sizes, from 5 decision variables and 5 constraints ($5 \times 5$) up to ($1000 \times 1000$). The results in Table 2 summarize the performance of all methods across different problem sizes. For a detailed analysis of constraint violation metrics, please refer to Appendix G. The RC and LT methods exhibit robust performance across the board, achieving objective values second only to EX while consistently maintaining low percentages of infeasible solutions and fast solution times across all problem sizes. These methods achieve several orders of magnitude speed-ups, scaling effectively even for large instances up to $1000 \times 1000$. The exact solver EX, while performing well on smaller problem sizes, fails to produce any solutions for instances of size $200 \times 200$ and larger within the 1000-second time limit, highlighting its limitations when handling more complex problems. N1, on the other hand, can find feasible solutions within a short time frame for smaller cases but suffers from severe numerical instability as the problem size increases. When scaled to $200 \times 200$, N1 also fails to produce a solution. The RR method, which relies on rounding relaxations, encounters significant feasibility challenges. Overall, this analysis

underscores that learning-based methods like RC and LT offer considerable advantages in both solution quality and computational speed, especially for large-scale problems, compared to exact solvers or other heuristics.

It is important to note that some of the Obj Mean and Median values are extremely large. This occurs when the baseline methods, such as EX and N1, generate poor-quality feasible solutions, particularly for larger problem instances. Since the decision variables are not explicitly upper/lower bounded, the baselines occasionally produce trivial yet suboptimal solutions, leading to inflated objective values. This issue is not limited to this particular case but also appears in other problem instances, further underscoring the limitations of the baseline methods in handling larger-scale optimization tasks effectively.

In addition to evaluating solution quality, feasibility, and solving/inference times, we also measured the offline training times for our two approaches on different problem sizes. These results, along with training times for other problem types, are presented in Appendix I, where it is evident that the training times for the learning-based methods scale well with problem size.

## 6.3 SIMPLE NON-CONVEX PROBLEM

To evaluate the performance on non-convex optimization tasks, we extended the convex quadratic programming problem by introducing a trigonometric term to the objective function, following the approach in Donti et al. (2021). This modification introduces non-convexity, increasing the challenge of finding optimal solutions. Additionally, we parameterized the constraint matrix to further enrich the complexity. Further details on the formulation, parameter generation, and experimental setup can be found in Appendix E. In addition, the scales of the problem and the experiment setting are also identical to those of the quadratic problems.

Table 3: Results for a Simple Non-convex Problem. See the caption of Table 2 for details. "% Unsolved" denotes the percentage of instances that could not be solved within the given time limit.

| Method | Metric | 5×5 | 10×10 | 20×20 | 50×50 | 100×100 | 200×200 | 500×500 | 1000×1000 |
|---|---|---|---|---|---|---|---|---|---|
| RC | Obj Mean | 0.855 | 1.236 | 0.598 | 1.352 | 1.208 | 1.522 | −0.351 | 4.051 |
| | Obj Median | 0.496 | 0.865 | 0.601 | 1.325 | 1.198 | 1.522 | −0.341 | 3.869 |
| | % Infeasible | 1% | 0% | 0% | 0% | 1% | 0% | 1% | 0% |
| | Time (Sec) | 0.0019 | 0.0019 | 0.0019 | 0.0020 | 0.0021 | 0.0022 | 0.0027 | 0.0049 |
| LT | Obj Mean | 0.778 | 1.175 | 0.608 | 0.932 | 1.627 | 0.729 | 0.254 | 6.651 |
| | Obj Median | 0.461 | 0.786 | 0.601 | 0.916 | 1.619 | 0.700 | 0.173 | 6.368 |
| | % Infeasible | 0% | 1% | 0% | 0% | 3% | 0% | 0% | 5% |
| | Time (Sec) | 0.0019 | 0.0019 | 0.0019 | 0.0020 | 0.0022 | 0.0022 | 0.0026 | 0.0054 |
| EX | Obj Mean | 0.001 | −0.182 | −0.453 | 1.649 | 256.926 | — | — | — |
| | Obj Median | −0.095 | −0.310 | −0.463 | −0.052 | 134.620 | — | — | — |
| | % Infeasible | 0% | 0% | 0% | 0% | 0% | — | — | — |
| | % Unsolved | 0% | 0% | 0% | 0% | 86% | 100% | 100% | 100% |
| | Time (Sec) | 0.1168 | 2.454 | 994.912 | 1001.52 | 1000.56 | — | — | — |
| RR | Obj Mean | −0.047 | −0.168 | −0.464 | −1.039 | −2.068 | −3.990 | −9.391 | — |
| | Obj Median | −0.089 | −0.325 | −0.476 | −1.215 | −2.307 | −4.327 | −9.221 | — |
| | % Infeasible | 64% | 86% | 97% | 100% | 100% | 100% | 100% | — |
| | % Unsolved | 0% | 0% | 0% | 0% | 0% | 0% | 0% | 100% |
| | Time (Sec) | 0.216 | 0.411 | 0.996 | 1.189 | 4.600 | 54.009 | 449.02 | — |
| N1 | Obj Mean | 1.690 | 1073.90 | 2.13e4 | 3.72e6 | 4411.45 | — | — | — |
| | Obj Median | 0.183 | 0.557 | 2.222 | 45.847 | 155.254 | — | — | — |
| | % Infeasible | 0% | 0% | 0% | 0% | 0% | — | — | — |
| | % Unsolved | 0% | 0% | 0% | 0% | 86% | 100% | 100% | 100% |
| | Time (Sec) | 0.040 | 0.103 | 0.144 | 8.968 | 940.43 | — | — | — |

The results presented in Table 3 reflect patterns similar to those observed in the quadratic problem. However, the sine function exacerbates the non-convexity of the problem, rendering it more challenging for traditional methods. Despite this added complexity, the RC and LT methods perform robustly, scaling to large instances for which the baselines fail to produce any solutions.

## 6.4 MULTI-DIMENSIONAL MIXED-INTEGER ROSENBROCK PROBLEM

The high-dimensional mixed-integer Rosenbrock problem is a challenging benchmark adapted from the classic Rosenbrock function, extended with integer variables, non-linear constraints, and parametric variations. It evaluates scalability and the ability to handle complex optimization landscapes. All parameters and the constraint structure, are described in Appendix E.

We conducted experiments on mixed-integer Rosenbrock problems with the number of decision variables ranging from 2 to 20,000; the number of constraints was fixed at 5. The results in Table 4 show that RC and LT exhibit strong performance, even outperforming EX in smaller cases. However, as the problem size increases to 10,000 variables, a noticeable decline in feasibility is observed for both RC and LT, while solver-based methods such as EX, N1, and RR fail to produce any solutions. As seen in previous experiments, RR, which relies on rounding relaxations, continues to suffer from significant infeasibility issues.

Table 4: Results for the Mixed-Integer Rosenbrock Problem. The number of decision variables varies from 2 to 20,000, while the number of constraints is 5. See the caption of Table 2 for details.

| Method | Metric | 2×4 | 20×4 | 200×4 | 2000×4 | 20000×4 |
|---|---|---|---|---|---|---|
| RC | Obj Mean / Median | 23.84/21.77 | 53.94/50.48 | 550.95/536.18 | 6034.64/5864.99 | 6.05e4/5.07e4 |
| | % Infeasible | 0% | 0% | 0% | 0% | 31% |
| | Time (Sec) | 0.0019 | 0.0019 | 0.0022 | 0.0032 | 0.0126 |
| LT | Obj Mean / Median | 23.40/21.00 | 54.97/52.34 | 563.67/541.81 | 5465.80/5400.32 | 1.86e5/1.74e5 |
| | % Infeasible | 0% | 0% | 0% | 0% | 45% |
| | Time (Sec) | 0.0020 | 0.0019 | 0.0022 | 0.0031 | 0.0119 |
| EX | Obj Mean / Median | 19.62/18.20 | 64.67/59.16 | 8.43e5/908.81 | 4.70e10/9262.09 | 1.09e15/1.03e5 |
| | % Infeasible | 0% | 0% | 0% | 0% | 0% |
| | % Unsolved | 0% | 0% | 0% | 4% | 22% |
| | Time (Sec) | 3.509 | 1004.61 | 1002.20 | 1001.69 | 1040.06 |
| RR | Obj Mean / Median | 22.24/22.19 | 12036.99/51.17 | 1.43e4/501.90 | 2.10e6/5436.60 | 1.75e8/7.03e6 |
| | % Infeasible | 45% | 41% | 18% | 1% | 4% |
| | % Unsolved | 0% | 0% | 42% | 93% | 78% |
| | Time (Sec) | 0.181 | 0.557 | 1.240 | 9.233 | 1064.12 |
| N1 | Obj Mean / Median | 40.37/27.93 | 87.83/77.34 | 3.72e8/957.42 | 8.27e12/9379.37 | 1.22e15/1.03e5 |
| | % Infeasible | 0% | 0% | 0% | 0% | 0% |
| | % Unsolved | 0% | 0% | 0% | 5% | 22% |
| | Time (Sec) | 0.032 | 0.081 | 0.261 | 71.911 | 782.052 |

## 6.5 EFFECT OF PENALTY WEIGHT

This section investigates the impact of the penalty weight, a critical hyperparameter, on the performance of the optimization methods. Experiments were conducted on three representative problems: a 1000×1000 convex quadratic problem, a 1000×1000 simple non-convex problem, and a 20000×4 Rosenbrock problem. For each problem, we evaluated the RC and LT methods under penalty weights of 1, 5, 10, 50, 100, 500, and 1000.

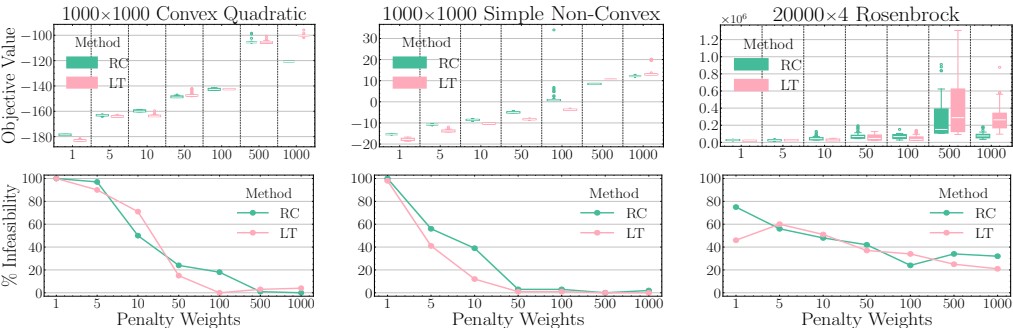

Figure 3: Illustration of the objective value (Top) and proportion of infeasible solutions (Bottom) on the test set. As the penalty weight increases, the fraction of infeasible solutions decreases while the objective value generally deteriorates, as expected.

Figure 3 reveals an inherent trade-off between achieving a higher proportion of feasible solutions and maintaining lower objective values. While increasing the penalty weight improves the feasibility rate, it often results in worse objective values. However, for the 20000×4 Rosenbrock problem, even with progressively increasing penalties, the predictor still yields many infeasible solutions. This limitation is addressed in Section 6.6.

## 6.6 EFFECT OF TRAINING SAMPLE SIZE

The large number of infeasible solutions observed in the 20000×4 Rosenbrock problem can primarily be attributed to significant overfitting within the model. Given that we have prior knowledge of the parameter distribution and our self-supervised learning approach does not rely on optimal solution labels, we can easily scale up the sample size to effectively mitigate overfitting.

To assess the impact of sample size, we trained the model on datasets of 800, 8,000, and 80,000 instances with 100 penalty weight, adjusting training epochs to 2000, 200, and 20 (with early stopping) to ensure comparable iterations. All other hyperparameters remained consistent to isolate the effect of sample size.

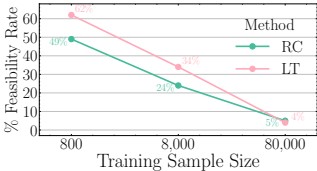 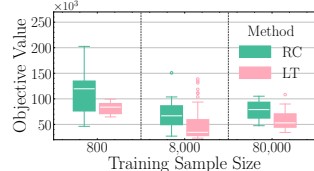

Figure 4: Illustration of the objective value (Left) and proportion of infeasible solutions (Right) of 20000×4 Rosenbrock problem on the test set. As the training sample size increases, the fraction of infeasible solutions decreases while the objective value generally deteriorates, as expected.

As shown in Section 6.6, increasing the sample size yields significant improvements in both objective values and feasibility. With 80,000 samples for training, the infeasibility ratio was reduced to 5% on the test set, demonstrating better generalization to unseen instances. This emphasizes the critical role of sufficient sample size and demonstrates the scalability advantage of our self-supervised framework.

## 7 CONCLUSION

We have introduced a new learning-based heuristic method for MINLP. Our approach includes two novel correction layers—rounding classification and learnable threshold—that enable neural networks to generate high-quality integer solutions while preserving gradient information for training through backpropagation. These layers allow us to tackle optimization tasks with discrete variables and non-linear constraints in a way that is scalable and computationally efficient. As a self-supervised approach, our method does not require collecting optimal solutions as labels, significantly reducing the time and effort typically needed for data collection.

Our experiments demonstrate that our learning-based methods outperform traditional solvers and other heuristics across various problem types, including convex quadratic, non-convex, and high-dimensional mixed-integer optimization problems. Despite the increasing complexity of these tasks, our methods maintain strong performance in terms of both feasibility and solution quality, particularly in high-dimensional settings where traditional approaches often fail to produce solutions within a reasonable time due to the curse of dimensionality. To our knowledge, our work is the first to tackle learning for parametric MINLPs in full generality.

Our method enables efficient heuristic solutions for large-scale parametric MINLPs, achieving better performance and computational efficiency, though feasibility is not guaranteed. Future work could explore improving feasibility through alternative constraint-handling techniques or post-processing. For certain problem classes, a subset of constraints could be relaxed into the loss function while directly optimizing over the rest using differentiable optimization layers Agrawal et al. (2019). Additionally, redesigning neural network architectures to handle varying instance parameters and decision variables is a promising direction, leveraging set-based, permutation-equivariant architectures such as graph neural networks Cappart et al. (2023); Dumouchelle et al. (2024); Chen et al. (2022b; 2024).

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

## A    GUMBEL-SIGMOID TRICK

The Gumbel-Sigmoid trick is a stochastic approximation method that enables gradient-based optimization for discrete variables. It introduces controlled random perturbations to the logits $h$, producing a continuous relaxation of the binary decision. The method relies on noise sampled from the Gumbel distribution, which is defined as:

$$g = -\log(-\log(U)), \quad U \sim \text{Uniform}(0, 1).$$

where $U$ is a random variable drawn from the uniform distribution over the interval $[0, 1]$. This noise $g$ is added to the logits to introduce randomness while preserving differentiability.

The Gumbel-Sigmoid function provides a soft, differentiable approximation of binary outputs as:

$$\text{Gumbel-Sigmoid}(h) = \frac{1}{1 + \exp\left(-\frac{h + g_1 - g_2}{\tau}\right)}$$

where $g_1$ and $g_2$ are independent samples from the Gumbel distribution, and $\tau > 0$ is the temperature parameter that controls the smoothness of the approximation. For large values of $\tau$, the output of the Gumbel-Sigmoid function is smooth and probabilistic, resembling a sigmoid function. As $tau \to 0$, the output approaches a hard binary decision, mimicking the behavior of a step function.

In optimization contexts, this property allows the Gumbel-Sigmoid trick to approximate discrete variables while maintaining differentiability, facilitating gradient-based methods. The noise introduced by $g_1$ and $g_2$ promotes exploration, potentially helping to escape poor local minima during training—an issue that is particularly common in discrete optimization problems.

In our experiments, we set $\tau = 1$ for simplicity. This setting has been empirically shown to perform well across various learning-to-optimize tasks but can be further tuned to balance the trade-off between exploration and exploitation.

## B    DETAILS OF CORRECTION LAYERS

The following subsections describe two distinct approaches for designing the correction layer $\varphi_{\Theta_2}$; the same network $\pi_{\Theta_1}$ is used in both approaches.

**Rounding Classification.**    Line 6 of Algorithm 1 is the key step in the *rounding classification* (RC) approach. For the integer variables, RC applies a stochastic soft-rounding to the output $\mathbf{h}_z^i$ of the neural network $\delta_{\Theta_2}(\bar{\mathbf{x}}^i, \boldsymbol{\xi}^i)$, yielding $\mathbf{b}^i \in \{0, 1\}^{n_z}$. An entry of $\mathbf{b}^i$ determines whether the continuously relaxed value $\bar{\mathbf{x}}_z^i$ of the corresponding variable is rounded down or up. In the backward pass, STE is used in line 4 for the rounding down operation. In line 6, the derivative of the Sigmoid function is used.

**Learnable Threshold.**    The key steps of the *learnable threshold* (LT) approach are described in lines 8 and 9 of Algorithm 1. Rather than use Gumbel-Sigmoid for the rounding as in RC, LT learns a vector of per-variable rounding thresholds, $\mathbf{v}^i \in [0, 1]^{n_z}$, that the Sigmoid generates in line 8 of Algorithm 1. A variable is rounded up if the fractional part of its relaxed value, $(\bar{\mathbf{x}}_z^i - \hat{\mathbf{x}}_z^i)$, exceeds the threshold. The indicator function $\mathbb{I}(\cdot)$ in line 9 produces a binary output in the forward pass. In the backward pass, the gradient is approximated by that of the Sigmoid function with a slope:

$$\mathbf{b}^i \leftarrow \frac{1}{1 + \exp\left(-10 \cdot (\bar{\mathbf{x}}_z^i - \hat{\mathbf{x}}_z^i - \mathbf{v}^i)\right)}.$$

Here, the slope is set to 10 to sharpen the Sigmoid function.

## C    EXAMPLE ILLUSTRATION

Figure 5 shows the evolution of both the relaxed and rounded solutions, $(\bar{x}, \bar{y})$ and $(\hat{x}, \hat{y})$, across different epochs of the training of an RC model on two-dimensional mixed-integer Rosenbrock

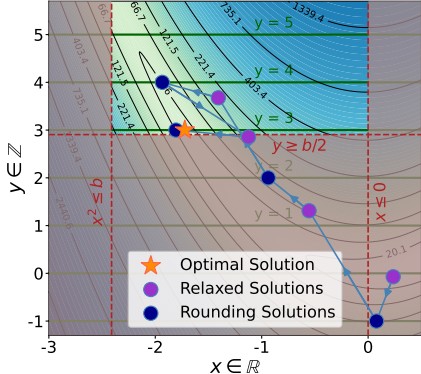

Figure 5: Example of the relaxed solutions $\bar{x}, \bar{y}$ and the rounding solutions $\hat{x}, \hat{y}$ across different epochs of training for the same sample instance using the Rounding Classification approach.

problems defined as follows:

$$\min_{x \in \mathbb{R}, y \in \mathbb{Z}} \quad (a - x)^2 + 50(y - x^2)^2$$

$$\text{subject to} \quad y \geq b/2, \quad x^2 \leq b, \quad x \leq 0, \quad y \geq 0.$$

In this formulation, $x$ is a continuous decision variable, and $y$ is an integer decision variable, subject to linear constraints. The instances have parameters $a$ and $b$, which represent the input features to the neural network; for the instance illustrated in Figure 5, these are set to $3.83$ and $6.04$, respectively.

The illustration shows that the training of this differentiable rounding approach converges remarkably well in this particular instance, with the final rounding solution being very close to the optimum. We will show this to be a generalizable phenomenon, with both of our learning approaches converging to highly accurate neural network models on a variety of problem classes and sizes.

## D  NEURAL NETWORK STRUCTURE AND HYPERPARAMETERS

The solution mapping $\pi_{\Theta_1}$ used across all learning-based methods—RC, LT, and RL—consists of five fully connected layers with ReLU activations. The rounding correction network $\varphi_{\Theta_2}$ for RC and LT is composed of four fully connected layers, also with ReLU activations, and incorporates Batch Normalization and Dropout with a rate of $0.2$ to prevent overfitting.

The hidden layer sizes were adjusted based on the problem size. For the convex quadratic and simple non-convex problems, the hidden layer width used in the learning-based methods was scaled accordingly, increasing from 16, 32, 64 up to 1024 for the corresponding problem sizes. Smaller problems, such as $5\times5$, used smaller hidden layers 16, while larger problems, such as $500\times500$, used hidden layers with widths up to 1024 to accommodate the complexity. Similarly, for the Rosenbrock problem, the hidden layer width was scaled based on the number of variables: a width of 4 was used for problems with 2 variables, 16 for problems with 20 variables, and up to 1024 for problems with $10,000$ variables.

The constraint penalty weight $\lambda$ was set to 300 for benchmark problems. All networks were trained using the AdamW optimizer with a learning rate of $10^{-3}$ and a batch size of 64 over 200 epochs. Early stopping was applied based on validation performance to ensure convergence without overfitting.

## E  MINLP PROBLEM SETUP AND PARAMETER SAMPLING

**Convex Quadratic Problems**  The convex quadratic problems used in our experiments are formulated as follows:

$$\min_{x \in \mathbb{Z}^n} \frac{1}{2} \boldsymbol{x}^\intercal \boldsymbol{Q} \boldsymbol{x} + \boldsymbol{p}^\intercal \boldsymbol{x} \text{ subject to } \boldsymbol{A} \boldsymbol{x} \leq \boldsymbol{b}$$

where the coefficients $\boldsymbol{Q} \in \mathbb{R}^{n \times n}$, $\boldsymbol{p} \in \mathbb{R}^n$, and $\boldsymbol{A} \in \mathbb{R}^{m \times n}$ were fixed, while $\boldsymbol{b} \in \mathbb{R}^m$ were treated as parametric coefficients (input features), varying across instances.

where $\boldsymbol{Q} \in \mathbb{R}^{n \times n}$ is a diagonal matrix with entries sampled uniformly from $[0, 0.01]$, ensuring convexity. The vector $\boldsymbol{p} \in \mathbb{R}^n$ has entries drawn from a uniform distribution over $[0, 0.1]$, while the constraint matrix $\boldsymbol{A} \in \mathbb{R}^{m \times n}$ is generated from a normal distribution with a standard deviation of $0.1$. The parameter $\boldsymbol{b} \in \mathbb{R}^m$, representing the right-hand side of the inequality constraints, is sampled uniformly from $[-1, 1]$. These variations in $\boldsymbol{b}$ across instances ensure the parametric nature of the problem.

**Simple Non-convex Problems**   The simple non-convex problem used in the experiments is derived by modifying the convex quadratic programming problem as follows:

$$\min_{x \in \mathbb{Z}^n} \ \frac{1}{2} \boldsymbol{x}^\intercal \boldsymbol{Q} \boldsymbol{x} + \boldsymbol{p}^\intercal \sin(\boldsymbol{x}) \text{ subject to } \boldsymbol{A}\boldsymbol{x} \leq \boldsymbol{b}$$

where the sine function is applied element-wise to the decision variables $\boldsymbol{x}$. This introduces non-convexity into the problem, making it more challenging compared to the convex case. For the simple non-convex problems, the coefficients $\boldsymbol{Q}$, $\boldsymbol{p}$, $\boldsymbol{A}$, and $\boldsymbol{b}$ are generated in the same way as in the quadratic formulation. However, an additional parameter $\boldsymbol{d} \in \mathbb{R}^m$ is introduced, with each element independently sampled from a uniform distribution over $[-0.5, 0.5]$. The parameter $\boldsymbol{d}$ modifies the constraint matrix $\boldsymbol{A}$ by adding $\boldsymbol{d}$ to its first column and subtracting $\boldsymbol{d}$ from its second column. Alongside $\boldsymbol{d}$, the right-hand side vector $\boldsymbol{b}$ remains a dynamic parameter in the problem.

**Ronsenbrock Problems.**   The mixed-integer Rosenbrock problem used in this study is defined as:

$$\min_{\boldsymbol{x} \in \mathbb{R}^n, \boldsymbol{y} \in \mathbb{Z}^n} \ (\boldsymbol{a} - \boldsymbol{x})^\intercal (\boldsymbol{a} - \boldsymbol{x}) + 50(\boldsymbol{y} - \boldsymbol{x}^2)^\intercal (\boldsymbol{y} - \boldsymbol{x}^2)$$

$$\text{subject to} \quad \|\boldsymbol{x}\|_2^2 \leq nb, \mathbf{1}^\intercal \boldsymbol{y} \geq \frac{nb}{2}, \boldsymbol{p}^\intercal \boldsymbol{x} \leq 0, \boldsymbol{q}^\intercal \boldsymbol{y} \leq 0,$$

where $\boldsymbol{x} \in \mathbb{R}^n$ are continuous decision variables and $\boldsymbol{y} \in \mathbb{Z}^n$ are integer decision variables. The vectors $\boldsymbol{p} \in \mathbb{R}^n$ and $\boldsymbol{q} \in \mathbb{R}^n$ are fixed for each instance, while the parameters $b$ and $\boldsymbol{a}$ vary. In details, the vectors $\boldsymbol{p} \in \mathbb{R}^n$ and $\boldsymbol{q} \in \mathbb{R}^n$ aregenerated from a standard normal distribution. The parameter $b$ is uniformly distributed over $[1, 8]$ for each instance, and the parameter $\mathbf{a} \in \mathbb{R}^n$ represents a vector where elements drawn independently from a uniform distribution over $[0.5, 4.5]$. The parameters $b$ and $\mathbf{a}$ influence the shape of the feasible region and the landscape of the objective function, serving as input features to the neural network.

# F   Ablation Study

**Overview.**   To better understand the contribution of the correction layers $\varphi_{\Theta_2}$, we include two ablation baselines in our experiments:

- **Rounding after Learning (RL)**: This baseline trains only the first neural network $\pi_{\Theta_1}$, which predicts relaxed solutions. Rounding to the nearest integer is applied post-training, meaning that the rounding step does not participate in the training process. This isolates the effect of excluding the corrective adjustments provided by $\varphi_{\Theta_2}$. This direct rounding can lead to significant deviations in the objective value and feasibility violations, underscoring the importance of end-to-end learning where updates are guided by the ultimate loss function.

- **Rounding with STE (RS):** In the Algorithm 2, continuous values predicted by $\pi_{\Theta_1}$ are rounded during training using the Straight-Through Estimator (STE), allowing gradients to pass through the rounding operator. While this mechanism applies a correction to produce integer values by rounding to the nearest integer, it is not learnable and does not adjust the rounding based on the parameter or the relaxation output. Thus, the correction is fixed and solely determined by the nearest-integer rounding, without leveraging additional learning for refinement.

---

**Algorithm 2** Rounding with STE for Learning-to-optimize MINLPs: Forward Pass.

---

**Require:** Training instance $\boldsymbol{\xi}^i$ and neural networks $\pi_{\Theta_1}(\cdot)$
 1: Predict a continuously relaxed solution $\bar{\mathbf{x}}^i \leftarrow \pi_{\Theta_1}(\boldsymbol{\xi}^i)$
 2: Round integer variables down: $\hat{\mathbf{x}}_z^i \leftarrow \lfloor \bar{\mathbf{x}}_z^i \rfloor$
 3: Compute $\mathbf{b}^i$ as the rounding direction using Gumbel-Sigmoid($\bar{\mathbf{x}}_z^i - \hat{\mathbf{x}}_z^i - 0.5$)
 4: Update integer variables: $\hat{\mathbf{x}}_z^i \leftarrow \hat{\mathbf{x}}_z^i + \mathbf{b}^i$
 5: **return** $\hat{\mathbf{x}}^i$

---

**Results and Insights.** The results of the ablation experiments, summarized in Table 5, Table 6 and Table 7, demonstrate the importance of the correction layers $\varphi_{\Theta_2}$ in improving both solution quality and feasibility. The experimental setup and model parameters used are consistent with those in the main text, ensuring the results are directly comparable. RL shows a significant drop in feasibility rates, highlighting the importance of incorporating learnable corrective adjustments during training. Similarly, while RS benefits from differentiability via STE, the lack of learnable correction limits its performance compared to RC and LT.

Table 5: Ablation Study for Convex Quadratic Problems. See the caption of Table 2 for details.

| Method | Metric | 5×5 | 10×10 | 20×20 | 50×50 | 100×100 | 200×200 | 500×500 | 1000×1000 |
|---|---|---|---|---|---|---|---|---|---|
| RL | Obj Mean | 0.563 | −2.182 | −4.569 | −13.732 | −15.985 | −37.363 | −86.385 | −165.047 |
| | Obj Median | 0.390 | −2.547 | −4.585 | −13.755 | −15.970 | −37.363 | −86.385 | −165.047 |
| | % Infeasible | 37% | 55% | 32% | 32% | 51% | 96% | 100% | 100% |
| | Time (Sec) | 0.0005 | 0.0006 | 0.0004 | 0.0004 | 0.0006 | 0.0005 | 0.0006 | 0.0011 |
| RS | Obj Mean | 1.087 | −0.694 | −3.202 | −11.213 | −9.240 | −21.975 | −47.732 | −92.846 |
| | Obj Median | 0.926 | −1.084 | −3.195 | −11.244 | −9.237 | −21.975 | −47.732 | −92.846 |
| | % Infeasible | 0% | 0% | 0% | 0% | 0% | 0% | 0% | 0% |
| | Time (Sec) | 0.0010 | 0.0013 | 0.0010 | 0.0012 | 0.0014 | 0.0012 | 0.0015 | 0.0034 |

Table 6: Ablation Study for Simple Non-Convex Problems. See the caption of Table 3 for details.

| Method | Metric | 5×5 | 10×10 | 20×20 | 50×50 | 100×100 | 200×200 | 500×500 | 1000×1000 |
|---|---|---|---|---|---|---|---|---|---|
| RL | Obj Mean | 0.374 | 0.741 | 0.054 | −0.421 | −0.915 | −2.177 | −10.684 | −21.837 |
| | Obj Median | 0.230 | 0.575 | 0.032 | −0.458 | −0.915 | −2.177 | −10.684 | −21.837 |
| | % Infeasible | 19% | 19% | 17% | 42% | 66% | 88% | 100% | 100% |
| | Time (Sec) | 0.0005 | 0.0005 | 0.0005 | 0.0005 | 0.0007 | 0.0005 | 0.0006 | 0.0012 |
| RS | Obj Mean | 0.866 | 1.421 | 0.588 | 2.704 | 3.531 | 7.038 | 15.065 | 37.559 |
| | Obj Median | 0.537 | 0.959 | 0.572 | 2.693 | 3.533 | 7.038 | 15.065 | 37.559 |
| | % Infeasible | 0% | 0% | 0% | 0% | 0% | 0% | 0% | 0% |
| | Time (Sec) | 0.0011 | 0.0010 | 0.0010 | 0.0011 | 0.0015 | 0.0014 | 0.0016 | 0.0032 |

Table 7: Ablation Study for Rosenbrock Problems. See the caption of Table 4 for details.

| Method | Metric | 2×4 | 20×4 | 200×4 | 2000×4 | 20000×4 |
|---|---|---|---|---|---|---|
| RL | Obj Mean / Median | 22.19/22.59 | 62.75/62.32 | 609.94/626.64 | 6131.90/5806.59 | 7.00e4/5.55e4 |
| | % Infeasible | 44% | 28% | 36% | 17% | 31% |
| | Time (Sec) | 0.0005 | 0.0005 | 0.0005 | 0.0006 | 0.0015 |
| RS | Obj Mean / Median | 25.26/25.98 | 67.36/65.05 | 667.72/641.89 | 7538.04/7597.98 | 8.41e4/8.44e4 |
| | % Infeasible | 0% | 3% | 0% | 1% | 30% |
| | Time (Sec) | 0.0010 | 0.0011 | 0.0011 | 0.0021 | 0.0091 |

## G DETAILS FOR CONSTRAINTS VIOLATIONS

In this section, we analyze constraint violations for three benchmark problems. The analysis focuses on both the frequency and magnitude of constraint violations, visualized through heatmaps for a comprehensive understanding. Each heatmap (Figure 6, Figure 7 and Figure 8) illustrates rows as test instances and columns as individual constraints. The heatmaps illustrate results for models trained on datasets of 8,000 samples, with penalty weights set to 200, 100, and 100, respectively, for the three problems.

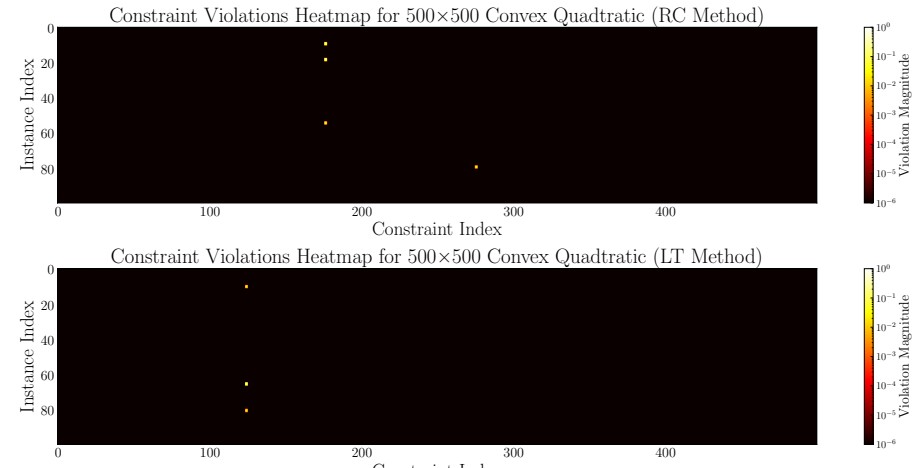

Figure 6: Illustration of Constraint Violation Heatmap for 500×500 Convex Quadratic Problem for RC method (Top) and LT method (bottom) on 100 test instances: Each row represents an instance in the test set, while each column corresponds to a specific constraint.

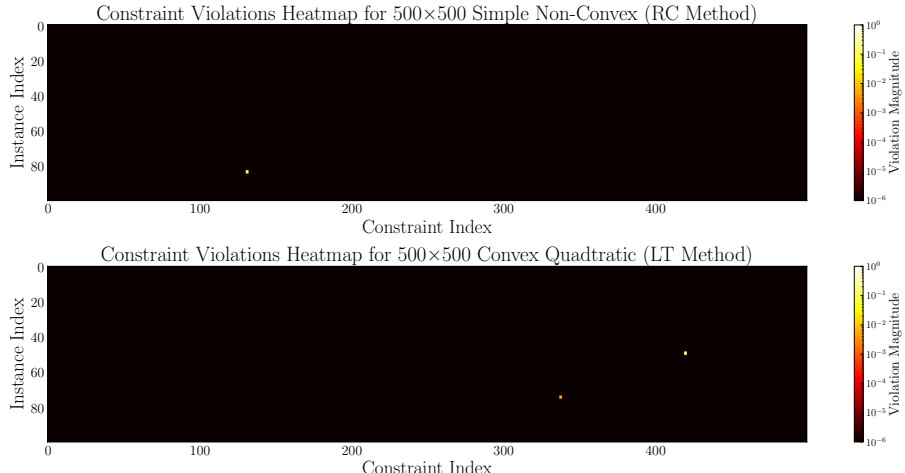

Figure 7: Illustration of Constraint Violation Heatmap for 500×500 Simple Non-Convex Problem for RC method (Top) and LT method (bottom) on 100 test instances: Each row represents an instance in the test set, while each column corresponds to a specific constraint.

The heatmap for the convex quadratic problem (Figure 6) and the simple non-convex problem (Figure 7) reveals a sparse distribution of violations, predominantly concentrated in a single constraint. This indicates that most constraints are consistently satisfied, with only a few isolated violations. Notably, 4 out of 100 test instances of convex quadratic problem under RC are infeasible, and among these, 3 violations occur within the same constraint, and all 3 infeasible solutions of convex quadratic from LT method also appear in the one constraint. Overall, the constraint violations are nearly negligible, confirming the effectiveness of the proposed methods.

Figure 8 highlights a much denser distribution of violations, reflecting the complexity of this benchmark. The nonlinear constraint $\|\mathbf{x}\|_2^2 \leq nb$ is particularly challenging, as shown by the more frequent and larger violations.

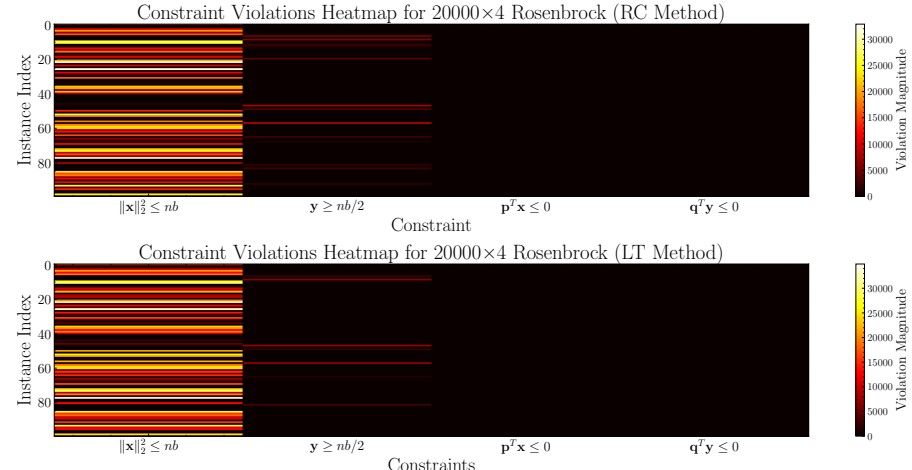

Figure 8: Illustration of Constraint Violation Heatmap for 20000×4 Rosenbrock Problem for RC method (Top) and LT method (bottom) on 100 test instances: Each row represents an instance in the test set, while each column corresponds to a specific constraint.

The heatmaps reveal key insights into the performance of the RC and LT methods. While the convex quadratic and simple non-convex problems exhibit minimal violations, the Rosenbrock problem highlights the difficulty of satisfying nonlinear constraints. These observations underscore the need for further refinement of penalty weights. Specifically, constraint-specific adjustments could mitigate violations by placing higher penalties on constraints that are harder to satisfy.

## H  EXPERIMENTS ON BINARY LINEAR PROGRAMS

**Dataset.**  For our experiments involving mixed-integer linear programs (MILPs), we utilized the 'Obj Series 1' dataset from the MIP Workshop 2023 Computational Competition (Bolusani et al., 2023). This dataset comprises 50 related MILP instances derived from a common mathematical formulation, where the instances differ in a subset of the objective function coefficients. Each instance contains 360 binary variables and 55 constraints, with 120 out of the 360 objective coefficients varying across instances. All other components of the problem remain consistent.

**Model Configuration.**  The neural network architecture and hyperparameters were consistent with those used for other experiments in the main paper. Specifically for the MILP problem in this study, the input dimension of the neural network was set to 120, corresponding to the number of varying objective function coefficients, and the output dimension was set to 360, representing the binary decision variables. The hidden layer consisted of 256 neurons.

**Results.**  Table 8 summarizes the results of the ILP experiments: Both learning-based methods (RC and LT) demonstrate the ability to generate high-quality feasible solutions efficiently, with RC even surpassing the heuristic-based method N1 in terms of objective value. However, N1 is the fastest method overall, showcasing the robustness and efficiency of the heuristic in the MILP solver. Notably, the training time for the learning-based models is approximately 120 seconds, making them well-suited for applications requiring repeated problem-solving.

## I  TRAINING TIME COMPARISON

In this section, we present the training times for the LR, LT, and RL methods across various problem sizes. All training runs were conducted using datasets of 9,000 instances for each problem with 1,000 instances reserved for validation per epoch. It is important to note that while the training

Table 8: Comparison of Optimization Methods on the MILP. See the caption of Table 2 for details.

| Method | Obj Mean | Obj Median | % Infeasible | Time (Sec) |
|--------|----------|------------|--------------|------------|
| RC | 9745.90 | 9763.00 | 0% | 0.04 |
| LT | 14149.00 | 14149.00 | 0% | 0.04 |
| EX | 8756.80 | 8747.00 | 0% | 28.91 |
| N1 | 11901.10 | 11933.00 | 0% | 0.01 |

process was set for 200 epochs, an early stopping strategy was applied, allowing the training to terminate earlier when performance plateaued.

Table 9: Training Times (in seconds) for LR, LT, and RL methods across different problem sizes for the Convex Quadratic Problem. Each method was set to train for 200 epochs, with early stopping applied.

| Method | 5x5 | 10x10 | 20x20 | 50x50 | 100x100 | 200x200 | 500x500 | 1000x1000 |
|--------|------|-------|-------|-------|---------|---------|---------|-----------|
| RC | 242.28 | 225.38 | 153.98 | 237.11 | 141.15 | 149.43 | 606.23 | 727.32 |
| LT | 217.01 | 225.38 | 154.33 | 158.61 | 128.86 | 139.17 | 458.62 | 462.41 |
| RL | 213.53 | 63.96 | 73.72 | 61.95 | 85.91 | 88.49 | 304.80 | 277.78 |

Table 10: Training Times (in seconds) for RC, LT, and RL methods across different problem sizes for the Simple Non-convex Problem. Each method was set to train for 200 epochs, with early stopping applied.

| Method | 5x5 | 10x10 | 20x20 | 50x50 | 100x100 | 200x200 | 500x500 | 1000x1000 |
|--------|------|-------|-------|-------|---------|---------|---------|-----------|
| RC | 257.28 | 144.46 | 173.02 | 138.53 | 136.01 | 104.05 | 116.01 | 156.85 |
| LT | 226.09 | 260.34 | 104.35 | 88.41 | 111.38 | 89.24 | 230.52 | 195.67 |
| RL | 111.07 | 75.67 | 79.28 | 58.86 | 81.43 | 84.28 | 149.87 | 131.42 |

Table 11: Training Times (in seconds) for RC, LT, and RL methods across different problem sizes for the Rosenbrock Problem. Each method was set to train for 200 epochs, with early stopping applied.

| Method | 2×4 | 20×4 | 200×4 | 2000×4 | 20000×4 |
|--------|------|------|-------|--------|---------|
| RC | 230.68 | 112.35 | 75.49 | 106.76 | 5227.05 |
| LT | 126.60 | 125.11 | 86.43 | 84.61 | 6508.41 |
| RL | 39.79 | 98.12 | 103.38 | 61.30 | 1920.59 |

Table 9, 10 and 11, summarize the training times (in seconds) required by each method for problems of different scales. These results highlight the computational efficiency of our methods during training, with training times for most problem instances remaining within a few hundred seconds. Even for large-scale problems, such as the 20000×4 Rosenbrock problem, the RC and LT method required only a few hours of training to handle a problem that exact solvers struggle to even find feasible solutions for in reasonable time. This efficiency is largely attributed to our simple neural network architecture, which enables scalable and efficient training. Thus, our method is particularly advantageous in real-world scenarios where rapid deployment and scalability are critical.

