# OpenReview forum: "Learning to Optimize for Mixed-Integer Nonlinear Programming"
_ICLR.cc/2025/Conference — Submitted to ICLR 2025_

### Official Review · Reviewer_refb · 2024-10-27

**Soundness:** 2
**Presentation:** 2
**Contribution:** 1
**Rating:** 3
**Confidence:** 4

**Summary:**

This paper proposes an end-to-end optimization method for solving general mixed-integer nonlinear programs (MINLPs). The proposed approach consists of two steps to generate solutions. In the first step, a neural network is employed to generate a relaxed solution that is close to the optimal solution. In the second step, another neural network provides update directions for continuous variables and rounding rules for integer variables. All of these neural networks are trained in a self-supervised manner. The Straight-Through Estimator is utilized to manage non-differentiable operations, such as rounding.

**Strengths:**

This paper focuses on applying machine learning methods to solve MINLPs.
- It proposes novel differentiable correction layers that can potentially handle the non-differentiability of integer outputs in deep learning models.

**Weaknesses:**

I have a few serious concerns below.

- First and foremost, since the proposed approach does not take advantage of the non-linear part, I think it could also be applicable to mixed-integer linear programs. Then why not also conduct computational experiments on those instances and show how good or bad it performs? We know that learning to solve MINLPs is rarely studied but being the first to address such a problem could be a trivial thing (not a significant contribution).

- Note that in the computational studies, only the right hand sides of constraints are perturbed, I recommend the authors perturb all parameters in the MINLP formulations and conduct experiments. The reason I ask such a question is, representing MINLPs using neural networks itself is a very important question (and challenging). Note that representing linear programs or mixed-integer linear programs via neural networks has theoretical foundations, see [1] [2]. Furthermore, I do not see equality constraints in the dataset.

- Can the authors consider more practical MINLP instances? Such as MINLPLIB (https://www.minlplib.org/). The dataset used in the manuscript is kind of like toy problems. I'm expecting to see the computational performances on real-life instances.

- The parameter $\lambda$ in loss function is an import hyper-parameter for balancing feasibility and optimality, and should be analyzed more carefully. Usually, penalty methods in L2O demonstrate very weak generalization capabilities. This kind of explains why the infeasibility ratio in Table 4 is so high. I do not think penalizing constraints in the loss function is a good way. Rather, the authors should design special algorithms to handle nonlinear (and possibly non-convex) constraints.

[1] Chen, Z., Liu, J., Wang, X., Lu, J. and Yin, W., 2022. On representing linear programs by graph neural networks. arXiv preprint arXiv:2209.12288.

[2] Chen, Z., Chen, X., Liu, J., Wang, X. and Yin, W., 2024. Expressive Power of Graph Neural Networks for (Mixed-Integer) Quadratic Programs. arXiv preprint arXiv:2406.05938.

**Questions:**

See the weakness part.

---

> ### Author Response · Authors · 2024-11-20
>
> Thank you for the detailed comments which we address next in this response as well as in the updated version of the submission.
>
> 1. **Experiments for MILP:**
>     - We agree and have already performed experiments on MILP problems. In Appendix H, we conducted additional experiments on MILPs using the `Obj Series 1` dataset from the MIP Workshop 2023 Computational Competition (https://github.com/ambros-gleixner/MIPcc23). The results demonstrate that our learning-based methods generate high-quality, feasible solutions efficiently, even outperforming the heuristic solution obtained at the root node in terms of objective value.  For this dataset, finding the optimal solution for these instances requires approximately 30 seconds, whereas the root node heuristics has an efficiency advantage (0.01 sec) over our methods (0.04 sec).
>
> 2. **Lack of significant contribution:**
>     - We respectfully disagree. The field of MINLP is large and growing, as evidenced by the focus attributed to MINLP in the solvers SCIP and Gurobi, both of which have been progressively expanding their capabilities to tackle non-convex MINLPs. This is likely due to practical applications requiring this capability. As you have noted, representing MINLPs is challenging. Our work takes a first step towards expanding the learning-to-optimize literature to MINLP by considering parametric versions of the problem. This allows us to focus on producing integer solutions, something that has not been tackled in prior work, even on MILP, and on constraint satisfaction. This is enabled not just by the loss function, but also the two-network architecture and the differentiable correction layers that are empirically very effective at generating feasible assignments to integer variables. For example, we have demonstrated the practical value of our approach through experiments on large-scale instances, such as the 200×200 convex quadratic problem and even larger instances, where traditional solvers and heuristic methods fail to find any feasible solution within reasonable time limits. In contrast, our method achieves feasible solutions with short training times and rapid inference speeds, often yielding high-quality results. If the reviewer is aware of other work that tackles these issues directly, we would appreciate some references.
>
> 3. **Only perturbed the RHS:**
>     - Thank you for those references, which we have commented on in the Related Work section of the updated paper.
>     - The mixed-integer Rosenbrock problem is parameterized by an n-dimensional vector $a$ in the objective as well as the scalar $b$ in the constraints.
>     - Additionally, in Section 6.2 of the updated paper, we have expanded the experiments to include perturbations to the constraint matrix A through an m-dimensional vector $d$, further broadening the scope of parameterization.
>
> 4. **Lack of equality constraints:**
>     - Generating a feasible equality constraint with integer variables can be highly non-trivial as it is at least as hard as the NP-Complete Subset-Sum problem. Generating a system of feasible equality constraints is even harder. In the absence of a reliable equality generation scheme, we opted to stick with inequality constraints, which are also extremely common in practice.

---

> > ### Author Response · Authors · 2024-11-20
> >
> > 5. **Real instances such as MINLPLib:**
> >     - These are individual instances from heterogeneous applications and often do not have a publicly available instance generation codebase for training datasets that we can use. The problems we look at are derived from prior work in learning-to-optimize or standard test functions for optimization (such as the Rosenbrock function).
> >
> > 6. **Need analysis on penalty weight:**
> >     - Thank you for this suggestion! As newly added in Section 6.5 and consistent with expectations, we observed improved constraint satisfaction at the expense of worse objective values as λ increases. This trade-off highlights the importance of carefully tuning penalty weight.
> >
> > 7. **High infeasibility ratio on Rosenbrock 20000x5:**
> >     - Since submitting the paper, we have addressed this issue; details are in the updated paper, Sections 6.5 and 6.6. In a nutshell, larger problems require larger penalty weights for constraint violations during training, which in turn could lead to overfitting in training instances. Simply increasing the size of the training set addresses this issue. As you can see in Figure 4 of the updated paper, the infeasibility rate on Rosenbrock 20000x5 is now 4%/5%, compared to the original 34%/24%.
> >
> > 8. **Alternatives to penalizing constraints in the loss function:**
> >     - We refer the reviewer to the standard textbook in continuous optimization by Nocedal and Wright [1] which, in Chapter 15 (“Fundamentals of Algorithms for Nonlinear Constrained Optimization”), categorizes penalty and augmented Lagrangian method (discussed in Chapter 17) as a key class of algorithms for constrained optimization. Other types of algorithms, such as interior-point methods, typically require second-order information to solve for KKT conditions, a requirement that would be prohibitive in the presence of a deep neural network. As such, we believe that the penalty method we use for training is suitable. This is complemented by strong empirical results and related literature on learning to optimize methods.
> >
> > [1] Nocedal, Jorge, and Stephen J. Wright, eds. Numerical optimization. New York, NY: Springer New York, 1999.

---

> > ### Comment · Reviewer_refb · 2024-11-25
> >
> > Thank the authors for your response. As I mentioned in my very first point, "**First and foremost, since the proposed approach does not take advantage of the non-linear part**", if the authors focus on L2O for MILPs, then compare your approach with the SOTA algorithms (there are so many baselines, and this is your focus). If your focus is on MINLPs, then **I do not see anything tailored for the non-linear part** and hence I do not see any significant contribution. I will maintain my score (reject).

---

> > > ### Author Response · Authors · 2024-11-27
> > >
> > > We appreciate the reviewer’s comments and their critical perspective. However, we respectfully disagree with the assertion that our contributions are limited by not tailoring the method for the non-linear part of MINLPs. (1) MINLPs are a general class of problems encompassing MILPs, and our work focuses on developing a scalable framework for mixed-integer optimization problems rather than limiting the scope to MILPs. (2) Regardless of the methodological preferences of reviewers, our work is the first L2O framework for MINLPs, demonstrating superior performance over solvers and heuristics. (3) While tailoring to non-linear structures is a valid approach, we believe it is not the only path to advancing MINLPs. Our method’s generality is a significant contribution.
> > >
> > > Thank you again for your critical insights, which helped refine our presentation.

---

> ### Comment · Area_Chair_HwFk · 2024-11-25
> **ICLR Public Discussion Phase Ending Soon**
>
> Dear Reviewer,
>
> This is a kind reminder that the dicussion phase will be ending soon on November 26th. Please read the author responses and engage in a constructive discussion with the authors.
>
> Thank you for your time and cooperation.
>
> Best,
>
> Area Chair

---

### Official Review · Reviewer_bMkZ · 2024-11-01

**Soundness:** 2
**Presentation:** 3
**Contribution:** 2
**Rating:** 3
**Confidence:** 3

**Summary:**

This paper addresses the challenging problem of Mixed-Integer Nonlinear Programming (MINLP) within a learning-to-optimize framework, a crucial area of research with significant applications across various domains. The integration of learning approaches into MINLPs is particularly complex due to the presence of integer decision variables, which complicates gradient-based optimization techniques. To tackle this issue, the authors propose two differentiable correction methods that enable neural networks to generate high-quality integer solutions while maintaining gradient information for backpropagation. Additionally, the authors conduct a comprehensive set of experiments to demonstrate the superiority of their proposed methods compared to traditional exact search algorithms and heuristic approaches.

**Strengths:**

1. MINLPs arise in numerous real-world applications, making the techniques proposed in this paper significantly relevant to practical problem-solving.

2. The paper is well-structured and clearly articulated, making it accessible to the reader.

3. The authors assert that they are the first to introduce Straight-Through Estimator (STE) and Gumbel-Sigmoid techniques in the context of learning-to-optimize, which they identify as pivotal for efficiently generating solutions to large-scale MINLP problems.

**Weaknesses:**

1. The fairness of the comparisons and the definitions used in the experiments are unconvincing for several reasons:

   - In lines 324 to 327, the authors list the solvers compared and the corresponding types of problems. However, they do not provide sufficient justification for the selection of these solvers or explain their relevance to the specific problem types addressed.

   - In lines 330 to 334, the authors mention modifications made to the original quadratic problems from Donti et al. (2021), but it remains unclear whether these modifications confer any advantages to the proposed method. Clarification is needed.

   - The metrics employed in the experiments raise concerns. For instance, while generating low percentages of infeasible solutions quickly is noted, the implications of this metric are questionable. The time required to convert an infeasible solution into a feasible one can be substantial, thus diminishing the significance of the reported speed.

   - In the experiments involving simple nonconvex problems, the use of the %Unsolved metric is unconventional. It is problematic to claim a problem is solved when the provided solution is still infeasible.

2. The loss function introduced in the paper essentially applies the Lagrangian multiplier method, which is not particularly novel in this field.

3. Additionally, there are several typographical errors throughout the paper. The authors should conduct a thorough proofreading before submission.

**Questions:**

1. In lines 330 to 334, the authors mention modifications made to the original quadratic problems from Donti et al. (2021). However, it remains unclear whether these modifications provide any advantages to the proposed method. A clarification on this point is necessary.

2. In Algorithm 1, the authors only consider a round-down direction for integer variables. It would be beneficial to explain why the round-up direction is excluded. If the round-up direction is relevant, this should be described in detail.

3. In the experiments, the authors allocate only a 60-second time budget to the exact solver. This limited timeframe may hinder the solver’s ability to find the optimal feasible solution, even if a few additional seconds are provided. It would be more informative to present a statistical distribution of % Infeasible versus Time (seconds) for the various methods evaluated.

---

> ### Author Response · Authors · 2024-11-20
>
> Thank you for the detailed comments which we address next in this response as well as in the updated version of the submission.
>
> 1. **Solver selection (Why Gurobi & SCIP?):**
>     - Both are generic MINLP solvers with some of the best-reported results according to independent benchmarking (see, for example, https://plato.asu.edu/ftp/minlp.html) as well as academic licenses for research. In fact, as noted by Lundell & Kronqvist (2022) (see [1] below for ref.), who performed a comprehensive benchmarking of more than ten MINLP solvers: “It is clear, however, that the global solvers Antigone, BARON, Couenne and SCIP are the most efficient at finding the correct primal solution when regarding the total time limit. [...] Gurobi also is very efficient when considering that it only supports a little over half of the total number of problems!”. As such, Gurobi and SCIP are very much representative of the state of MINLP solving.
>
> 2. **Problem selection (Why change Donti et al’s QP):**
>     - We describe this instance generation process in more detail in Appendix E of the updated paper. Simply put, Donti et al. studied continuous quadratic problems whereas we are interested in their discrete counterparts. The modifications do not advantage our methods at all. Specifically, we added integrality constraints, which inherently make the problems more challenging, and removed equality constraints to avoid infeasible instances in our discrete setting. These changes were necessary to adapt the problems to the discrete domain, but they do not alter the problem in a way that benefits our proposed methods.
>
> 3. **Feasibility issue:**
>     - Indeed, the neural network output may not be feasible, though our training loss function encourages the model to produce feasible solutions, and our empirical results show rather low infeasibility rates. We have expanded the analysis of infeasibility and show that with sufficiently large penalty weights in training and large enough training data, integer-feasible solutions can be generated most of the time; we refer to Section 6.5 and Section 6.6 of the updated paper.
>    - Guaranteeing a feasible solution for a MINLP is NP-Hard in general. As such, no heuristic algorithm, ML-based or not, can be guaranteed to produce feasible solutions in polynomial time. However, our ML model output can be passed on to an exact MINLP solver, such as Gurobi/SCIP, which can then attempt to construct a fully feasible solution. In Gurobi, this can be done using “[variable hints](https://docs.gurobi.com/projects/optimizer/en/current/reference/attributes/variable.html#varhintval)”.
>     - We agree that including a more explicit evaluation of post-processing time and its impact on overall efficiency would strengthen the paper. However, our results already demonstrate that our method provides an efficient, practical, and scalable solution, especially for large-scale instances where solvers fail to find any feasible solutions for each instance within a reasonable time.
>
> 4. **Metrics (%Infeasible and %Unsolved):**
>     - Due to numerical issues, it is possible for a MINLP algorithm to produce a solution that is thought to be feasible when it actually is not. We perform this check and record the “% Infeasible" rate. As for the “%Unsolved” metric: Given that MINLPs are NP-Hard to solve, no polynomial-time algorithm is guaranteed to generate a feasible solution in a bounded amount of time. Tracking the number of test instances for which no solution is generated by a method is thus important. We believe that these metrics, in conjunction with objective function value mean/median as well as running time, provide a complete picture of method performance.
>     - In our updated experiments, with access to improved computational resources, the simple non-convex problem does not currently exhibit partial unsolved cases within the time limit—i.e., either all instances are solved, or none are. However, for the Rosenbrock problem, we still observe and report the “%Unsolved” metric.
>
> 5. **Loss function novelty:**
>     - Indeed, penalizing constraint violations in an objective function is a standard technique for continuous constrained optimization. However, this approach has not been used at all to learn to generate solutions for mixed-integer non-linear programs. Our proposed methods are the first to address learning-to-optimize in the context of general parametric MINLPs. This is enabled not just by the loss function but also by integer outputs from the two-network architecture and the differentiable correction layers that are empirically very effective at generating feasible assignments to integer variables.
>
> 6. **Typos:**
>     - We have corrected every typo we could find in the updated version of the submission.
>
> [1] Lundell, Andreas, and Jan Kronqvist. "Polyhedral approximation strategies for nonconvex mixed-integer nonlinear programming in SHOT." Journal of Global Optimization 82.4 (2022): 863-896.

---

> > ### Author Response · Authors · 2024-11-20
> >
> > And there are our responses to the questions:
> >
> > 8. **Modifications of  Donti et al’s QP**:
> >     - See the response #2.
> >
> > 9. **Why round-down instead of round-up:**
> >     - It is possible to round up instead. We don’t believe this will affect the learning process in any meaningful way. Our proposed algorithm separates variables into their integer and fractional components, with the fractional part being allocated to 0 or 1 through the correction layer. Rounding down is a reasonable operation to isolate the integer part of the variable.
> >
> > 10. **60-second solving time is limited:**
> >     - We have performed additional experiments using a time limit of 1000 seconds. The results for the convex quadratic problem are presented in Section 6.2 of the updated submission. Our methods still outperform baselines by a substantial margin. For problems larger than 200×200, exact solvers often fail to find a feasible solution within 1000 seconds—or even hours or days—highlighting the challenges of scaling traditional methods. For the simple non-convex problem (Section 6.3) and the Rosenbrock problem (Section 6.4), experiments with a 1000-second time limit are currently underway, and we will update the manuscript with these results as soon as they are complete.
> >     - We do note, however, that a short time limit of 60 seconds is also appropriate for real-time solution generation, which is common to many applications. While increasing the solver time limit may improve the quality of exact solutions, our approach offers a critical advantage: the ability to produce solutions in milliseconds. This efficiency makes our method well-suited for applications requiring real-time or near-real-time decision-making, where longer solver runtimes are impractical.
> >
> > 11. **Experiments for % Infeasible versus time:**
> >     - We acknowledge that providing a statistical distribution of infeasibility over time could offer further insights into solver performance. However, conducting such an experiment would be extremely time-consuming, especially for large-scale problems. For example, finding an optimal or even feasible solution for a single instance of a large-scale problem, such as a 1000×1000 quadratic problem or the 20000×4 Rosenbrock problem, could take several days or even longer, depending on the complexity of the instance and the computational resources available. This makes a full statistical analysis of infeasibility versus time impractical for large-scale instances.
> >     - To address this concern, we have included in Section 6.1 (Figure 2) a record of the solver's performance over time on smaller-scale problems. The figure illustrates how the objective value evolves as the solver progresses, showing that it takes several hundred seconds for the solver to achieve a feasible solution comparable to the ones generated by our method. Even in these smaller-scale settings, our method demonstrates a clear efficiency advantage by providing high-quality, feasible solutions in milliseconds. This efficiency becomes even more critical in large-scale or real-time applications where extended computational time is impractical.

---

> ### Comment · Area_Chair_HwFk · 2024-11-25
> **ICLR Public Discussion Phase Ending Soon**
>
> Dear Reviewer,
>
> This is a kind reminder that the dicussion phase will be ending soon on November 26th. Please read the author responses and engage in a constructive discussion with the authors.
>
> Thank you for your time and cooperation.
>
> Best,
>
> Area Chair

---

### Official Review · Reviewer_Gthk · 2024-11-01

**Soundness:** 3
**Presentation:** 3
**Contribution:** 2
**Rating:** 5
**Confidence:** 4

**Summary:**

This paper proposes two differential correction layers (rounding classification and learnable threshold) that generate integer outputs while preserving gradient information.  The experiments demonstrate that the proposed learning-based approach consistently produces high-quality solutions.

**Strengths:**

- The topic of MINLP is an interesting and important topic in the field of learning to optimize.
- This paper combines the gradient information during the optimization.
- The presentation in this paper is good.

**Weaknesses:**

- The STE is not novel in the ML field. Moreover, the author may want to explain that combining the gradient information cannot lead to local optima.
- While many works on learning to optimize use GNN to process problems with different sizes, the proposed method seems to use MLP with fixed-size inputs. Thus, the network may fail to process problems of various sizes.
- The author may investigate the effects of different $\lambda$ on the performance.
- The author may conduct experiments on more complex instances, and the 60-second time limit is too short. Existing works in learning to optimize conduct experiments on challenging instances with at least 1000 sec of time limit [1,2].

[1]  A GNN-Guided Predict-and-Search Framework for Mixed-Integer Linear Programming

[2] GNN&GBDT-Guided Fast Optimizing Framework for Large-scale Integer Programming

**Questions:**

- Could you please explain how the proposed method handles problems with different sizes?
- Could the proposed method generalize to large instances, such as those with thousands of constraints and variables?

---

> ### Author Response · Authors · 2024-11-20
>
> Thank you for the thoughtful comments, which we will address next in this response as well as in the updated version of the submission.
>
> 1. **STE is not novel, and gradient cannot lead to local optima:**
>     - We would appreciate a clarification on what you mean by this comment about the gradient that cannot lead to local optima so we can address it appropriately. Thank you!
>     - We note that we use STE in two novel integer correction layers that we introduce in this paper. We don't claim novelty for the STE itself. These layers build upon the differentiability offered by STE and adaptively adjust the rounding direction. In the newly included ablation study (Appendix F), we explicitly evaluate a baseline method that uses STE alone to round values to the nearest integer (Rounding with STE, RS), and its performance is limited compared to our method.
>
> 2. **MLP can only process fixed-size inputs vs. GNN in prior work:**
>     - GNN models for mixed-integer linear programs are suitable because a MILP can be represented exactly by a bipartite variable-constraint graph over which the GNN operates. The same does not hold for our MINLPs, as they can have non-linear constraints. In the MILP case, an edge between a variable and a constraint represents that the variable has a non-zero coefficient in that constraint. The same trick cannot be used for MINLP representation, making a direct application of GNN impossible. Future work on this front would be interesting, though it is orthogonal to what we are proposing here.
>     - While our method does not directly adapt to varying problem sizes, its self-supervised nature avoids the need for optimal solutions as labels, allowing us to efficiently train models tailored for large-scale problems without relying on generalization across sizes. For instance, it is practical to train separate models for different problem sizes at a low cost. In contrast, many previous GNN-based methods [1, 2] rely on optimal solutions as labels, which is impractical for large-scale problems to get sufficient labels. This makes scalability and generalization to large-scale problems even more critical. We demonstrate this capability by training on 20000×4 Rosenbrock problems (Sections 6.4, 6.6) and additional experiments on 1000×1000 problems (Sections 6.2, 6.3).
>
> 3. **Effects of different penalty weights on the performance:**
>     - As newly added in Section 6.5, and consistent with expectations, we observed improved constraint satisfaction at the expense of worse objective values as λ increases. This trade-off highlights the importance of carefully tuning penalty weight.
>
> 4. **1000 sec of time limit for solver:**
>     - We have performed additional experiments using a time limit of 1000 seconds. The results for the convex quadratic problem are presented in Section 6.2 of the updated submission. Our methods still outperform baselines by a substantial margin. For problems larger than 200×200, exact solvers often fail to find a feasible solution within 1000 seconds—or even hours or days—highlighting the challenges of scaling traditional methods. For the simple non-convex problem (Section 6.3) and the Rosenbrock problem (Section 6.4), experiments with a 1000-second time limit are currently underway, and we will update the manuscript with these results as soon as they are complete.
>     - We do note, however, that a short time limit of 60 seconds is also appropriate for real-time solution generation, which is common to many applications. While increasing the solver time limit may improve the quality of exact solutions, our approach offers a critical advantage: the ability to produce solutions in milliseconds. This efficiency makes our method well-suited for applications requiring real-time or near-real-time decision-making, where longer solver runtimes are impractical.
>
> 5. **Larger Instances**:
>     - We have included experiments with 1000×1000 instances in Sections 6.2 and 6.3. In the context of MINLP, these problem dimensions are already considered highly challenging. For instance, with hundreds of variables and constraints, exact solvers such as Gurobi and SCIP often require prohibitively long computation times to produce feasible or near-optimal solutions.
>     - It is important to note that these problem sizes are significantly larger than most benchmark problems (e.g., 100×100) used in the current learning-to-optimize literature for continuous cases [3,4].
>
>
> [1] A GNN-Guided Predict-and-Search Framework for Mixed-Integer Linear Programming
>
> [2] GNN&GBDT-Guided Fast Optimizing Framework for Large-scale Integer Programming
>
> [3] DC3: A learning method for optimization with hard constraints
>
> [4] Self-Supervised Primal-Dual Learning for Constrained Optimization

---

> ### Comment · Area_Chair_HwFk · 2024-11-25
> **ICLR Public Discussion Phase Ending Soon**
>
> Dear Reviewer,
>
> This is a kind reminder that the dicussion phase will be ending soon on November 26th. Please read the author responses and engage in a constructive discussion with the authors.
>
> Thank you for your time and cooperation.
>
> Best,
>
> Area Chair

---

### Official Review · Reviewer_rKoe · 2024-11-04

**Soundness:** 3
**Presentation:** 3
**Contribution:** 2
**Rating:** 6
**Confidence:** 4

**Summary:**

The paper proposes an end-to-end method for learning solutions of integers programs by enabling differentiation through the rounding operation within model training. This is done by using the Straight-through Estimator (STE) combined with the Gumbel-noise method, which smooths the discrete function representing the rounding operations to obtain useful gradients for backpropagation. The paper provides a comprehensive evaluations of the proposed method across several optimization tasks.

**Strengths:**

The paper is well written and organized. The idea of integrating the rounding operations within model training is sound and allows to obtain superior models with respect to Learning to Optimize models who solve a relaxed version and perform the rounding operations at inference time, as shown in the experimental sections. Computational advantages are also significant with respect to traditional numerical solver.

**Weaknesses:**

My main concern is that the proposed method cannot ensure constraint satisfactions, since it uses a soft constraint approach. I believe that integer variables also makes difficult to perform projections to restore feasibility at inference time. Nonetheless, the percentage of infeasible solution generated by the proposed method is low, and the results shown in the Table 5 suggest that using a Lagrangian-inspired method might allow to obtain a better estimate of the dual variables, which might help to reduce constraint violations.
The paper might benefit from a more systematic evaluation of the impact of different constraint functions to the feasibility/violations produced by the proposed method, which might allow to identify scenarios and pattern where the proposed method (does not) produce constraint violation.

**Questions:**

Could you please expand on the constrain violations produced by your method? Do you have an understanding of how the proposed method handle different constraint functions? For instance, what type of constraint are well-handled? What, instead, are more difficult to satisfy?

---

> ### Author Response · Authors · 2024-11-20
>
> Thank you for the thoughtful comments, which we will address next in this response as well as in the updated version of the submission.
>
> 1. **Infeasibility handling:**
>     - Indeed, the neural network output may not be feasible, though our training loss function encourages the model to produce feasible solutions, and our empirical results show rather low infeasibility rates. We have expanded the analysis of infeasibility and show that with sufficiently large penalty weights in training and large enough training data, integer-feasible solutions can be generated most of the time; we refer to Section 6.5 and Section 6.6 of the updated paper. Although the feasibility is not guaranteed, our results already demonstrate that our method provides an efficient, practical, and scalable solution, especially for large-scale instances (e.g. , 1000×1000 quadratic problem and the 20000×4 Rosenbrock problem) where solvers fail to find any feasible solutions for each instance within a reasonable time.
>     - In addition, guaranteeing a feasible solution for a MINLP is NP-Hard in general. As such, no heuristic algorithm, ML-based or not, can be guaranteed to produce feasible solutions in polynomial time. However, our ML model output can be passed on to an exact MINLP solver, such as Gurobi/SCIP, which can then attempt to construct a fully feasible solution. In Gurobi, this can be done using “[variable hints](https://docs.gurobi.com/projects/optimizer/en/current/reference/attributes/variable.html#varhintval)” .
>
> 2. **Constraints analysis and penalty weights tuning:**
>     - As newly added in Section 6.5 and consistent with expectations, we observed improved constraint satisfaction at the expense of worse objective values as λ increases. This trade-off highlights the importance of carefully tuning penalty weight.
>     - Additionally, we have expanded our analysis of constraint violations, which is now presented as heatmaps across various benchmark problems in updated Appendix G. Violations in the convex quadratic problem and the simple non-convex problem, are rare and generally minor in magnitude, as illustrated in Figures 6 and 7. When violations do occur, they are concentrated on a single (identical) constraint and affect only a small number of instances. This indicates that most constraints in these problem types are well-handled by our method. In contrast, for the Rosenbrock problem at an extreme scale (20000×4), violations are more frequent and substantial, particularly for the nonlinear constraint $\|\| \mathbf{x} \|\|_2^2 \leq n b$, as shown in Figure 10. By the way, this feasibility issue is significantly mitigated with more sampling data (in Section 6.6).
>     - Building on this analysis, we agree with the reviewer that dynamically analyzing and tuning the penalty weights for specific constraints based on their scale and difficulty during training could be a promising approach to improving feasibility rates for challenging constraints. We thank the reviewer for this valuable suggestion, which opens up an exciting direction for future research.

---

> > ### Comment · Reviewer_rKoe · 2024-11-25
> > **Response to Authors' comments**
> >
> > I would like to thank the authors for the additional experiments and in general for working on improving the paper during the discussion period. I think that the analysis on constraint violations is useful to both the readers, which can have a better sense of how the method handle the constraint functions, and to the authors, which can develop further intuitions to refine the proposed method. Despite that, I still think that constraint satisfaction is a key challenge in L2O setting, which is only partially addressed in this work, and as such I would like to keep my score.

---

> ### Comment · Area_Chair_HwFk · 2024-11-25
> **ICLR Public Discussion Phase Ending Soon**
>
> Dear Reviewer,
>
> This is a kind reminder that the dicussion phase will be ending soon on November 26th. Please read the author responses and engage in a constructive discussion with the authors.
>
> Thank you for your time and cooperation.
>
> Best,
>
> Area Chair

---

> ### Author Response · Authors · 2024-11-27
>
> We thank the reviewer for their insightful suggestion. Regarding constraint satisfaction, we would like to clarify that it remains a well-known open challenge in the L2O setting, even for simpler cases like MILPs. For example, in L2O for MILP [1,2], infeasibility is also a persistent issue, and part of the contribution lies in reducing its occurrence. Additionally, we note that submission 7722, a concurrent work submitted to ICLR, reports feasibility rates of 50.8%, 97.1%, and 99.4% for MILP problems. It is, therefore, very challenging to expect perfect feasibility from a solver-free approach like ours.
>
> To our knowledge, no solver-free approach guarantees feasibility for general constrained discrete problems, and typical strategies involve either refining the solver's search space to reduce infeasibility or using infeasible solutions as starting points for solvers to repair feasibility.
>
> We believe our work makes a meaningful contribution by significantly lowering the infeasibility rate while maintaining solver-free operation and scalable performance. We thank the reviewer again for their feedback, which encourages us to continue refining and presenting our methods.
>
> [1] Solving Mixed Integer Programs Using Neural Networks
>
> [2] Contrastive Predict-and-Search for Mixed Integer Linear Programs

---

> > ### Comment · Reviewer_rKoe · 2024-11-27
> > **Response to Authors' comments**
> >
> > I agree that constraint satisfaction is an open challenge in L2O setting and I think that this work has actually taken a step forward in the advancement of L2O for MILPs. Frankly, my opinion on this paper is mildly positive, but there seems to be very contrastive views of this paper among reviewers.

---

### Author Response · Authors · 2024-11-21
**A Comment to the Reviewers to Clarify the Contribution and Announce the Updates**

We appreciate the reviewer’s time and effort in providing detailed feedback. We would like to take this opportunity to clarify the contributions and highlight the updates made to strengthen the paper based on the reviewer’s suggestions.

**Contribution:**

1. **Efficient Learning-to-Optimize (L2O) method for parametric mixed-integer nonlinear programming (MINLP).** We propose a novel two-network architecture tailored for the challenging domain of general parametric MINLPs. This simple yet efficient end-to-end framework can be trained within a few hundred seconds offline and provide extremely fast inference in milliseconds using standard hardware. We demonstrate that the proposed method produces high-quality solutions with a strong feasibility rate for large-scale instances, even for problems where traditional solvers (including SOTA commercial solvers) struggle to find any feasible solution in a reasonable time.

2. **Self-Supervised Learning without the need for labeled data generated by classical solvers.** The proposed architecture can be effectively trained with first-order gradient solvers in a self-supervised setting using the Lagrangian loss function without requiring optimal solutions as labels. This self-supervised approach avoids the computational overhead of collecting large-scale labeled data from classical numerical solvers such as Gurobi or SCIP. This makes our framework particularly practical and scalable for solving high-dimensional parametric MINLPs as we demonstrate in our extensive experimental case study section.

**Updates made to the manuscript:**

1. **1000-second time limit (Section 6.2):** We conducted additional experiments with a 1000-second time limit for convex quadratic problems. These results demonstrate that our methods still outperform baselines by a substantial margin. For the simple non-convex problem (Section 6.3) and the Rosenbrock problem (Section 6.4), experiments with a 1000-second time limit are currently underway. Once complete, these results will also be included in the manuscript.

2. **Larger instances (Section 6.2 and 6.3):** We performed experiments on larger problem instances, including 1000×1000 convex quadratic problems and 1000×1000 non-convex problems, i.e., problems with 1000 decision variables and 1000 constraints. These scales illustrate the robustness and scalability of our approach. It is important to note that these problem sizes are significantly larger than most benchmark problems (e.g., 100×100) used in the current learning-to-optimize literature for continuous cases [1,2].

3. **Penalty weight analysis (Section 6.5):** We conducted a detailed analysis of the effect of penalty weights on solution quality and constraint satisfaction. As expected, increasing the penalty weights improves feasibility rates but may slightly degrade objective values. This trade-off underscores the importance of tuning penalty weights carefully.

4. **Sample size analysis (Section 6.6):** We examined the impact of varying training sample sizes on model performance for 20000×4 Rosenbrock. Larger training datasets significantly reduced infeasibility rates and improved generalization to unseen instances. Notably, since our method is self-supervised and does not require optimal solutions as labels, the cost of increasing the training sample size is remarkably low, making this approach both practical and efficient for large-scale problems.

5.  **Ablation studies (Appendix F):** We added ablation studies to isolate and evaluate different aspects of our method, such as the impact of the end-to-end learning and the differentiable integer correction, to better understand their contribution to overall performance.

6. **Constraint violation analysis (Appendix G):** We analyzed constraint violations in terms of both frequency and magnitude across three benchmark problems. This provides a detailed understanding of how well our method satisfies constraints.

7. **MILP experiments (Appendix H):** We have conducted additional experiments for MILP from the MIP Workshop 2023 Computational Competition to evaluate the generalizability of our methods.

[1] DC3: A learning method for optimization with hard constraints

[2] Self-Supervised Primal-Dual Learning for Constrained Optimization

---

> ### Author Response · Authors · 2024-11-26
> **Updated Results: Enhanced Comparison with Solvers under a 1000-Second Time Limit**
>
> We would like to inform the reviewers and readers that we have updated the manuscript to update experiments for the Simple Non-Convex Problem (Section 6.3) and the Rosenbrock Problem (Section 6.4) with a 1000-second time limit for solvers. **All solver experiments are now performed under this 1000-second time limit for consistency.**
>
> 1. Convex Quadratic Problem:
>     - Our method consistently finds solutions within 0.005 seconds with strong feasibility rates across all problem sizes.
>     - Solvers, in contrast, fail to find feasible solutions within 1000 seconds for the 200×200, 500×500, and 1000×1000 problem sizes.
>     - Starting from a 10×10 problem size, our method significantly outperforms heuristics on root node
>
> 2. Simple Non-Convex Problem:
>     - Our method achieves solutions within 0.005 seconds with strong feasibility rates across all problem sizes.
>     - Solvers exhibit 86% failure rates in finding feasible solutions within 1000 seconds for the 100×100 problem size and fail entirely for the 200×200, 500×500, and 1000×1000 problem sizes.
>     - Starting from a 10×10 problem size, our method significantly outperforms heuristics on root node.
>
> 3. Multi-Dim Rosenbrock Problem:
>     - For all instances except the largest (20000×4), our method finds feasible solutions in 0.003 seconds or less.
>     - For a 2000×4 problem size, 4% of instances can not be solved in 1000 seconds, while for a problem size of 20000×4, this percentage rises to %22.
>     - When the solver finds feasible solutions within 1000 seconds, it performs poorly, even for small instances. For example, at the 20×4 problem size, solutions from solvers are significantly worse than those generated by our method.
>     - For the largest instance (20000×4), increasing the number of samples during training and adjusting penalty weights can greatly improve the feasibility of our method.
>
> These results further emphasize the efficiency, scalability, and practicality of our approach, especially in scenarios where solvers struggle to find feasible solutions or deliver competitive performance within reasonable time limits. We deeply appreciate your ongoing engagement with our work, and we hope these updates further clarify the contributions and practical significance of our method.

---

### Meta-Review · Area_Chair_HwFk · 2024-12-19

**Metareview:**

This paper proposes a gradient-based learning method to generate integer solutions for mixed-integer nonlinear programs (MINLPs). To address the challenge of handling non-differentiable operations in predicting integer variables, the paper introduces two differentiable correction layers—rounding classification and learnable thresholds—that provide useful gradients for backpropagation. Experiments demonstrate the method’s effectiveness in producing high-quality solutions.

However, the reviewers have pointed out some important weaknesses. First, the proposed method primarily combines existing differentiable techniques, which limits the contribution of the paper. Second, more insights are needed on improving constraint satisfaction, especially for large-scale instances with nonlinear constraints. Third, the paper should evaluate more practical instances with large scales or diverse constraints to provide a more comprehensive understanding of the method's performance. Therefore, I recommend the next version of this paper to incorporate more insights and experiments.

**Additional Comments On Reviewer Discussion:**

Reviewers rKoe, Gthk, bMkz, and refb rated this paper as 6: borderline accept (keep the score), 5: borderline reject (keep the score), 3: reject (keep the score), and 3: reject (keep the score), respectively.

The reviewers raised the following concerns.
- Novelty and Contribution (raised by Reviewers Gthk, bMkz, and refb)
- Constraint satisfaction (raised by Reviewers rKoe and refb)
- Insufficient experiments (raised by Reviewers Gthk and refb)
- Scalability (raised by Reviewers Gthk)
- Unclear experiment details (raised by Reviewers bMkZ)

The authors have addressed several reviewers' concerns by providing additional experiments on larger instances, analysis for sample sizes and constraint violations, and explanations of experiment details. However, some fatal weaknesses have not been properly addressed by the authors' rebuttal. First, the proposed method primarily combines existing differentiable techniques, which limits the contribution of the paper. Second, more insights are needed on improving constraint satisfaction, especially for large-scale instances with nonlinear constraints. Third, the paper should evaluate more practical instances with large scales or diverse constraints to provide a more comprehensive understanding of the method's performance.

Therefore, I will not recommend accepting this paper in its current state.

---

### Decision · Program_Chairs · 2025-01-22

Reject